# microRNA-33 maintains adaptive thermogenesis via enhanced sympathetic nerve activity

Takahiro Horie [1✉], Tetsushi Nakao[1], Yui Miyasaka[1], Tomohiro Nishino [1], Shigenobu Matsumura [2], Fumiko Nakazeki [1], Yuya Ide[1], Masahiro Kimura[1], Shuhei Tsuji [1], Randolph Ruiz Rodriguez[1], Toshimitsu Watanabe[1], Tomohiro Yamasaki[1], Sijia Xu[1], Chiharu Otani[1], Sawa Miyagawa[1], Kazuki Matsushita[1], Naoya Sowa[1], Aoi Omori[1], Jin Tanaka[2], Chika Nishimura[3], Masataka Nishiga [1], Yasuhide Kuwabara [1], Osamu Baba[1], Shin Watanabe[1], Hitoo Nishi[1], Yasuhiro Nakashima[1], Marina R. Picciotto [4], Haruhisa Inoue [5,6,7], Dai Watanabe[3], Kazuhiro Nakamura [8], Tsutomu Sasaki [9], Takeshi Kimura[1] & Koh Ono [1✉]

Adaptive thermogenesis is essential for survival, and therefore is tightly regulated by a central neural circuit. Here, we show that microRNA (miR)-33 in the brain is indispensable for adaptive thermogenesis. Cold stress increases miR-33 levels in the hypothalamus and miR-33$^{-/-}$ mice are unable to maintain body temperature in cold environments due to reduced sympathetic nerve activity and impaired brown adipose tissue (BAT) thermogenesis. Analysis of miR-33$^{f/f}$ dopamine-β-hydroxylase (*DBH*)-Cre mice indicates the importance of miR-33 in *Dbh*-positive cells. Mechanistically, miR-33 deficiency upregulates gamma-aminobutyric acid (GABA)$_A$ receptor subunit genes such as *Gabrb2* and *Gabra4*. Knock-down of these genes in *Dbh*-positive neurons rescues the impaired cold-induced thermogenesis in miR-33$^{f/f}$ *DBH*-Cre mice. Conversely, increased gene dosage of miR-33 in mice enhances thermogenesis. Thus, miR-33 in the brain contributes to maintenance of BAT thermogenesis and whole-body metabolism via enhanced sympathetic nerve tone through suppressing GABAergic inhibitory neurotransmission. This miR-33-mediated neural mechanism may serve as a physiological adaptive defense mechanism for several stresses including cold stress.

[1] Department of Cardiovascular Medicine, Graduate School of Medicine, Kyoto University, Kyoto, Japan. [2] Laboratory of Physiological Functions of Food, Division of Food Science and Biotechnology, Graduate School of Agriculture, Kyoto University, Kyoto, Japan. [3] Department of Biological Sciences, Graduate School of Medicine, Kyoto University, Kyoto, Japan. [4] Department of Psychiatry and Interdepartmental Neuroscience Program, Yale University School of Medicine, New Haven, CT, USA. [5] Center for iPS Cell Research and Application (CiRA), Kyoto University, Kyoto, Japan. [6] iPSC-based Drug Discovery and Development Team, RIKEN BioResource Research Center (BRC), Kyoto, Japan. [7] Medical-risk Avoidance based on iPS Cells Team, RIKEN Center for Advanced Intelligence Project (AIP), Kyoto, Japan. [8] Department of Integrative Physiology, Nagoya University Graduate School of Medicine, Nagoya, Japan. [9] Laboratory of Nutrition Chemistry, Division of Food Science and Biotechnology, Graduate School of Agriculture, Kyoto University, Kyoto, Japan. ✉email: thorie@kuhp.kyoto-u.ac.jp; kohono@kuhp.kyoto-u.ac.jp

The balance between energy intake and expenditure determines energy homeostasis, and an imbalance between the two can contribute to obesity. Worldwide, the problem of obesity is becoming increasingly apparent, given the range of obesity-related disorders such as cardiovascular diseases, metabolic diseases, and cancer. According to estimates from the World Health Organization (WHO), in 2016, more than 1.9 billion adults were overweight, and of these over 650 million were obese. If current trends continue unabated, more than 1 billion individuals, or 20% of the world's population, are predicted to be obese by 2030[1]. In spite of these dramatic increases in obesity in recent years, therapeutic approaches have mainly been limited to recommendations regarding lifestyle modifications, like caloric restriction or exercise regimes. However, the development of therapies beyond these approaches requires a deeper understanding of the fundamental mechanisms underlying obesity.

Brown adipose tissue (BAT) is one of the two main tissues that comprise the adipose organ, along with white adipose tissue. Its primary function is to maintain body temperature in response to cold stress by utilizing fatty acids for heat generation, in a process known as cold-induced adaptive thermogenesis[2]. BAT is enriched with mitochondria and derives its thermogenesis ability mainly from uncoupling protein 1 (UCP1), which uncouples substrate oxidation from ATP production into heat production in mitochondria. First extensively investigated in rodents, the identification of BAT in adult humans has accelerated the pace of research into BAT biology in recent years[3–5]. Because of its ability to dissipate excess energy into heat to maintain energy balance, understanding BAT and its activation may open up promising approaches for the prevention and treatment of obesity. Adaptive thermogenesis is essential for survival, and thus it is tightly regulated by a central neural circuit mechanism[6]. BAT is highly innervated by sympathetic ganglionic neurons, which are innervated by sympathetic preganglionic neurons at the intermediolateral nucleus (IML) of the spinal cord. Sympathetic premotor neurons in the rostral raphe pallidus nucleus (rRPa) of the medulla, which innervate to preganglionic neurons in the IML, are highly regulated by the balance between glutamatergic excitatory and gamma-aminobutyric acid (GABA)-ergic inhibitory inputs from the upper neurons in the hypothalamic regions like the dorsomedial hypothalamic nucleus (DMH) and the preoptic area (POA)[6,7]. Although this neural circuit is gradually becoming more clarified through experimental studies, it has yet to be fully elucidated.

MicroRNAs (miRNAs; miRs) are a class of short, noncoding RNA that substantially affect many biological pathways via post-transcriptional repression of their target genes[8]. Some miRNAs are involved in the biogenesis and function of BAT, and these may be potential targets for the treatment of obesity[9]. We and other groups have shown that miR-33, transcribed from the intron of sterol regulatory element-binding factor (SREBF) 2, regulates lipid homeostasis with their host genes by targeting the ATP-binding cassette subfamily A1 (ABCA1), which mediates cholesterol export to apolipoprotein A-I[10–12]. Inhibition or deletion of miR-33 in mice (miR-33$^{-/-}$) was found to reduce the formation of atherosclerosis or aortic aneurysms by improved serum cholesterol profile, enhanced cholesterol reverse transport, and ameliorated inflammation[13–15]. Although rodents have only one miR-33 in an intron of the Srebf2 gene, humans have another, miR-33b, in an intron of the SREBF1 gene. In studies with humanized miR-33b knock-in (miR-33b$^{+/+}$) mice, miR-33b was found to accelerate atherosclerosis[16,17]. In addition, inhibition of miR-33 has been shown to improve dyslipidemia in non-human primates[18,19]. Thus, miR-33a/b has been suggested as a promising target for the treatment of dyslipidemia and atherosclerosis. However, miR-33$^{-/-}$ mice unexpectedly exhibited an obese phenotype when fed a high-fat diet (HFD)[20], a result that was further confirmed by other miR-33 inhibition studies using an antisense oligonucleotide[21] and another miR-33-deleted mouse line[22]. Taken together, these findings suggest that miR-33 plays a substantial role in adipocyte biology, particularly in the onset of obesity.

In the current study, we analyzed several miR-33 genetically modified mice and found that miR-33 maintains BAT thermogenesis. Unexpectedly, we found that this effect was caused by altered sympathetic nerve activity. Our analysis of miR-33$^{f/f}$ DBH-Cre mice indicated that dopamine-β-hydroxylase (Dbh)-positive cells were responsible for reduced sympathetic nerve activity and decreased thermogenesis in miR-33$^{-/-}$ mice. Mechanistically, we propose that miR-33 deficiency upregulates the expression of GABA$_A$ receptor subunit genes such as Gabrb2 and Gabra4, which could reduce sympathetic nerve activity and BAT thermogenesis. In support of this hypothesis, we found that adeno-associated virus serotype 9 (AAV9)-mediated conditional knockdown of these genes in the brain of miR-33$^{f/f}$ DBH-Cre mice reversed the decrease in BAT thermogenesis. Conversely, miR-33b$^{+/+}$ mice displayed a phenotype opposite that of miR-33$^{-/-}$ mice. These data reveal a neural mechanism for thermogenesis via miR-33, which exerts sympathetic nerve activation and BAT thermogenesis in response to cold stress by suppressing GABAergic inhibitory neurotransmission in the brain.

## Results

**miR-33$^{-/-}$ mice become obese with reduced oxygen consumption when fed an HFD**. We and others have previously reported that miR-33$^{-/-}$ mice become obese after HFD feeding[20,22], and as expected, in the present study miR-33$^{-/-}$ mice gained much more weight than littermate miR-33$^{+/+}$ control mice when given a 45% HFD feeding for 12 weeks (Supplementary Fig. 1a). Food intake was slightly but significantly higher in miR-33$^{-/-}$ mice on the HFD, as reported previously (Supplementary Fig. 1b). We measured fat tissue weight and found that BAT was significantly heavier in miR-33$^{-/-}$ mice than in miR-33$^{+/+}$ control mice after 12 weeks of HFD feeding (Supplementary Fig. 1c). Histological analysis showed that there was much more white adipose tissue (WAT)-like lipid droplets in the BAT of miR-33$^{-/-}$ mice (Supplementary Fig. 1d). Therefore, we speculated that BAT function might be impaired in miR-33$^{-/-}$ mice fed an HFD. We measured oxygen consumption rate and found that it was significantly lower in HFD-fed miR-33$^{-/-}$ mice (Supplementary Fig. 1e, f). Gene expression levels of thermogenic genes including Ucp1 in the BAT were not significantly reduced in these mice (Supplementary Fig. 1g). However, it is known that long-term HFD feeding can affect several metabolic components, and these in turn may influence each other through compensatory mechanisms. To understand the direct effects of HFD feeding, we conducted a shorter-term study lasting 2 weeks, during which mice were fed a diet with higher fat content (60%) to increase the burden for a short time. While mice gained weight on the 60% HFD after 2 weeks, the changes in body weight were similar between miR-33$^{+/+}$ and miR-33$^{-/-}$ mice (Supplementary Fig. 2a). Under this condition, the expression levels of thermogenic genes were significantly reduced in the BAT of miR-33$^{-/-}$ mice (Supplementary Fig. 2b). Increased protein levels of UCP1 were induced by feeding with an HFD in the BAT of miR-33$^{+/+}$ mice, while this induction was not observed in the BAT of miR-33$^{-/-}$ mice (Supplementary Fig. 2c, d). Histological examination of the BAT showed larger lipid droplets and attenuated UCP1 induction in miR-33$^{-/-}$ mice

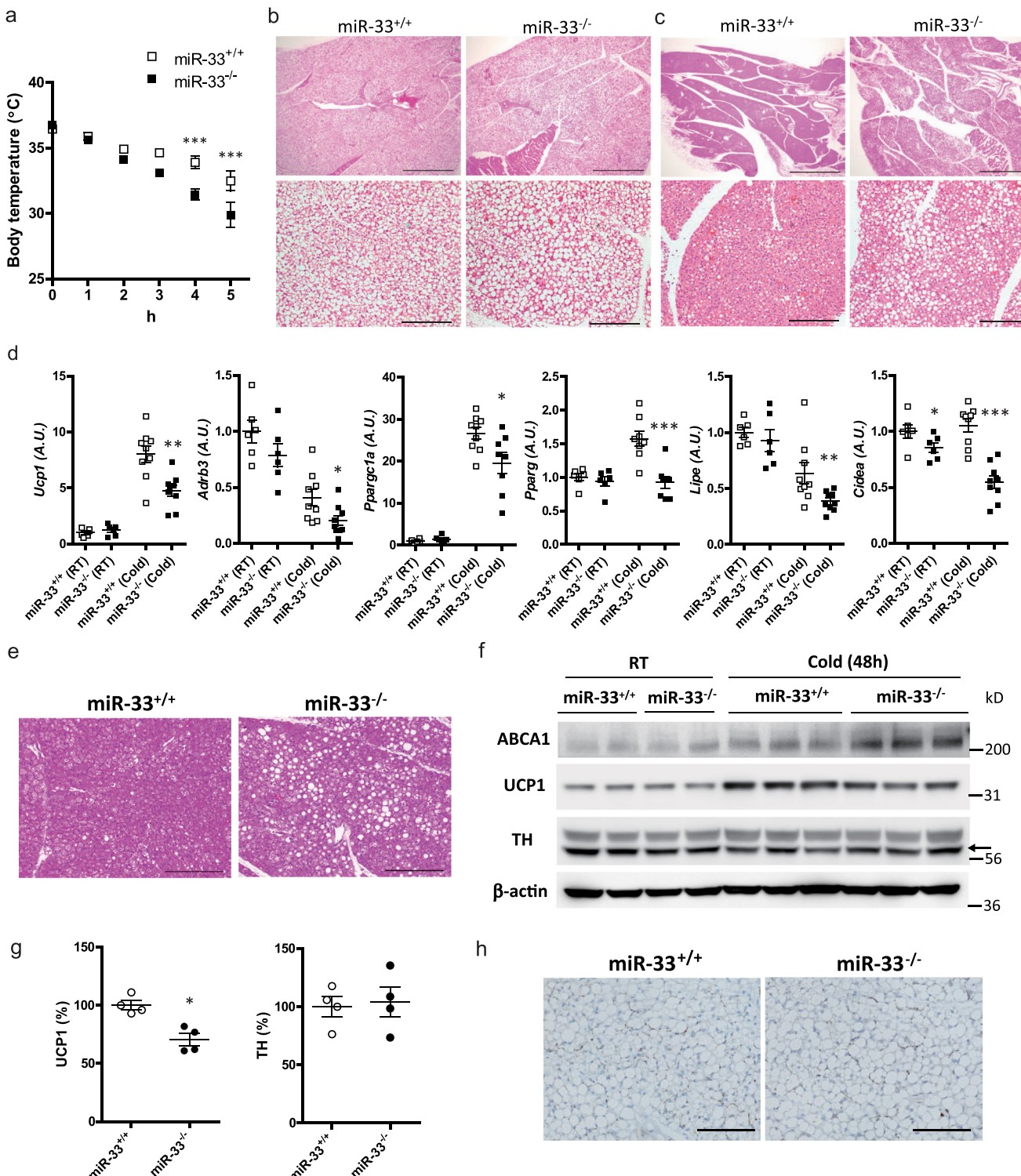

(Supplementary Fig. 2e–g). These data indicate that BAT activation and function are impaired in miR-33$^{-/-}$ mice when fed an HFD, which may contribute to the increased obesity of these mice.

**miR-33$^{-/-}$ mice exhibit impaired thermogenesis via reduced BAT activation.** Because HFD-induced UCP1 increase was attenuated in the BAT of miR-33$^{-/-}$ mice and diet-induced thermogenesis is known to be one form of adaptive thermogenesis[23], we speculated that cold-induced adaptive thermogenesis may also be

impaired in miR-33$^{-/-}$ mice. To evaluate cold-induced adaptive thermogenesis, we placed 8-week-old mice in a 4 °C cold environment and measured serial core body temperature. We found that rectal temperature decreased faster in miR-33$^{-/-}$ mice than in miR-33$^{+/+}$ mice (Fig. 1a). Lipid droplets in the BAT were slightly but significantly larger in miR-33$^{-/-}$ mice than miR-33$^{+/+}$ mice at room temperature, which is mildly cold for mice (Fig. 1b and Supplementary Fig. 3a). After deep cold exposure (4 °C) for 6 h, lipid droplets were dramatically reduced in miR-33$^{+/+}$ mice but were still present in miR-33$^{-/-}$ mice (Fig. 1c and Supplementary Fig. 3a).

**Fig. 1 miR-33$^{-/-}$ mice exhibit impaired thermogenesis and BAT activation in a cold environment. a** Serial core body temperature change of miR-33$^{+/+}$ and miR-33$^{-/-}$ mice kept at 4 °C. $n = 10$ and 11 mice per group, ***$p < 0.001$, two-way repeated-measures ANOVA with Bonferroni's post hoc test. **b** Representative images of HE staining for BAT in miR-33$^{+/+}$ and miR-33$^{-/-}$ mice kept at room temperature (RT). $n = 6$ mice per group. Scale bar, 1 mm (upper), 200 μm (lower). **c** Representative images of HE staining for BAT in miR-33$^{+/+}$ and miR-33$^{-/-}$ mice kept at 4 °C for 6 h. $n = 5$ mice per group. Scale bar, 1 mm (upper), 200 μm (lower). **d** Quantitative PCR analysis of thermogenic genes in BAT of miR-33$^{+/+}$ and miR-33$^{-/-}$ mice kept at room temperature (RT) and 4 °C for 6 h (cold). $n = 6$ mice per group at room temperature, $n = 8$–9 mice per group at 4 °C for 6 h. ***$p < 0.001$, **$p < 0.01$, *$p < 0.05$, two-sided Mann–Whitney test. **e** Representative images of HE staining for BAT of miR-33$^{+/+}$ and miR-33$^{-/-}$ mice kept at 4 °C for 48 h. $n = 6$ mice per group. Scale bar, 200 μm. **f** Western blotting analysis of ABCA1, UCP1, TH, and β-actin in the BAT of miR-33$^{+/+}$ and miR-33$^{-/-}$ mice kept at room temperature (RT) and 4 °C for 48 h (cold). $n = 2$, 3 mice per group. The arrow indicates the specific bands for TH staining. **g** Densitometry for UCP1 and TH in the BAT of miR-33$^{+/+}$ and miR-33$^{-/-}$ mice kept at 4 °C for 48 h. $n = 4$ mice per group, *$p < 0.05$, two-sided Mann–Whitney test. **h** Representative images of immunohistochemistry for TH in the BAT of miR-33$^{+/+}$ and miR-33$^{-/-}$ mice at room temperature. $n = 3$ mice per group. Scale bar, 100 μm. All data are presented as mean ± SEM.

These data suggest that miR-33$^{-/-}$ mice cannot effectively use lipids as fuel. The expression levels of thermogenic genes in the BAT were similar between miR-33$^{+/+}$ and miR-33$^{-/-}$ mice at room temperature (Fig. 1d). After cold exposure, *Ucp1*, *Ppargc1a*, and *Pparg* were induced in miR-33$^{+/+}$ mice, but these inductions were attenuated in miR-33$^{-/-}$ mice. Other genes such as *Adrb3*, *Lipe*, and *Cidea* were also decreased in miR-33$^{-/-}$ mice after cold exposure.

We further checked the protein levels of several genes in the BAT after 48 h of cold exposure. Under this condition, more lipid droplets were still observed in miR-33$^{-/-}$ mice (Fig. 1e, Supplementary Fig. 3b, c). We found that ABCA1 transporter, one of the well-known target genes of miR-33, was up-regulated in miR-33$^{-/-}$ mice (Fig. 1f), and that UCP1 was induced after cold exposure in miR-33$^{+/+}$ mice, with this induction attenuated in miR-33$^{-/-}$ mice (Fig. 1f, g). This was also confirmed by immunostaining (Supplementary Fig. 3d). Sympathetic nerve innervation into BAT was similar between miR-33$^{+/+}$ and miR-33$^{-/-}$ mice, which was confirmed by western blotting and immunohistochemistry for tyrosine hydroxylase (TH) (Fig. 1f–h). The oxygen consumption rate at 23 °C (room temperature) was slightly but significantly reduced in miR-33$^{-/-}$ mice in the dark phase. (Fig. 2a, c). Carbon dioxide production was also significantly reduced in miR-33$^{-/-}$ mice in the dark phase (Fig. 2b, d). The respiratory exchange rate (RER) and locomotor activity were similar for both miR-33$^{+/+}$ and miR-33$^{-/-}$ mice (Fig. 2e, f). After 2 days of cold exposure at 18 °C, the difference in oxygen consumption became evident and significantly reduced in miR-33$^{-/-}$ mice in the dark phase (Fig. 2g, i). The carbon dioxide production was also significantly reduced in miR-33$^{-/-}$ mice (Fig. 2h, j). The RER was significantly higher in miR-33$^{-/-}$ mice in the light phase, whereas the locomotor activity was not different between miR-33$^{+/+}$ and miR-33$^{-/-}$ mice in this condition (Fig. 2k, l). Fatty acid oxidation (FAO) activity in the BAT was significantly reduced in miR-33$^{-/-}$ mice in a cold environment (Supplementary Fig. 3e). Oxygen consumption rate in a thermoneutral zone (30 °C) is shown in Supplementary Fig. 3f, g. We measured the maximum oxygen consumption rate using norepinephrine (NE) subcutaneous injection after three days of cold exposure at 18 °C[24,25]. NE-induced maximum oxygen consumption was significantly reduced in miR-33$^{-/-}$ mice (Fig. 2m). Moreover, we measured both BAT and rectal temperature at the same time in a cold condition. BAT temperature declined more slowly than rectal temperature, which indicates that BAT thermogenesis was occurring. The decline in BAT temperature was more prominent and the difference in BAT and rectal temperature was smaller in miR-33$^{-/-}$ mice than miR-33$^{+/+}$ mice (Fig. 2n). These data indicate that miR-33$^{-/-}$ mice exhibit impaired adaptive thermogenesis with reduced BAT activation in a cold environment.

**The abundance of miR-33 in the human and mouse brain**. We reported the abundant expression of miR-33 in the brain of mice[26]. Although rodents have only one miR-33 in an intron of the *Srebf2* gene, humans have another, miR-33b, in an intron of the *SREBF1* gene. We checked the expression profiles of miR-33a and miR-33b in various organs using a commercially available human RNA panel. Consequently, miR-33a and miR-33b were most abundant in the brain compared with other organs, which was accompanied by the expression of their host genes, *SREBF2* and *SREBF1*, respectively (Supplementary Fig. 4a). We previously generated humanized miR-33b knock-in mice (miR-33b$^{+/+}$ mice), in which the human miR-33b sequence is inserted within the same intron of *Srebf1* as in human[16]. In these mice, miR-33b is physiologically co-expressed with its host gene, *Srebf1*. The expressions of *Srebf2*, *Srebf1*, miR-33a, and miR-33b in several regions of the central nervous system in miR-33b$^{+/+}$ mice are shown in Supplementary Fig. 4b, indicating that miR-33b$^{+/+}$ mice can be used as a gain-of-function model of miR-33.

**miR-33$^{-/-}$ mice exhibit reduced sympathetic nerve activity**. Because BAT activation is mainly under the control of sympathetic nerves[27] and miR-33 is abundantly expressed in the central nervous system, we hypothesized that the reduced BAT activation observed in miR-33$^{-/-}$ mice might be caused by differences in sympathetic nerve activity. To assess this hypothesis, we surgically denervated the nerves that innervate interscapular BAT (iBAT), as reported previously[28]. In each mouse, the left side of BAT was denervated and the right side of BAT was sham-operated on. Successful denervation was confirmed by measurement of NE content in the BAT (Supplementary Fig. 5a). Histological analysis showed enlarged lipid droplets in the denervated BAT, indicating that lipids could not be utilized effectively by denervated tissue (Supplementary Fig. 5b). One week after the operation, miR-33$^{+/+}$ and miR-33$^{-/-}$ mice were placed in a 4 °C environment for 5 h. Reduced expression levels of thermogenic genes including *Ucp1* were observed in miR-33$^{-/-}$ mice on the sham-operated side; however, these differences disappeared in the denervated side of the BAT (Fig. 3a). Expression levels of several internal control genes including *B2m*, *36b4*, and *Gapdh* are shown in Supplementary Fig. 5c. These results suggest that sympathetic nerves are responsible for the different expression levels of thermogenic genes between miR-33$^{+/+}$ and miR-33$^{-/-}$ mice in response to cold exposure.

Next, we measured NE turnover (NETO) in BAT using α-methyl-DL tyrosine (AMPT), an inhibitor of TH that is a rate-limiting enzyme for NE synthesis, with known usage as a potential surrogate for sympathetic nerve activity[25,28]. NETO in BAT was significantly lower in miR-33$^{-/-}$ mice, indicating that sympathetic nerve activity related to BAT was suppressed in miR-33$^{-/-}$ mice (Fig. 3b). Furthermore, we counted the number of c-fos-positive cells, which can be used as an indicator of neuronal activation, in the rRPa of the medulla (Fig. 3c). rRPa harbors

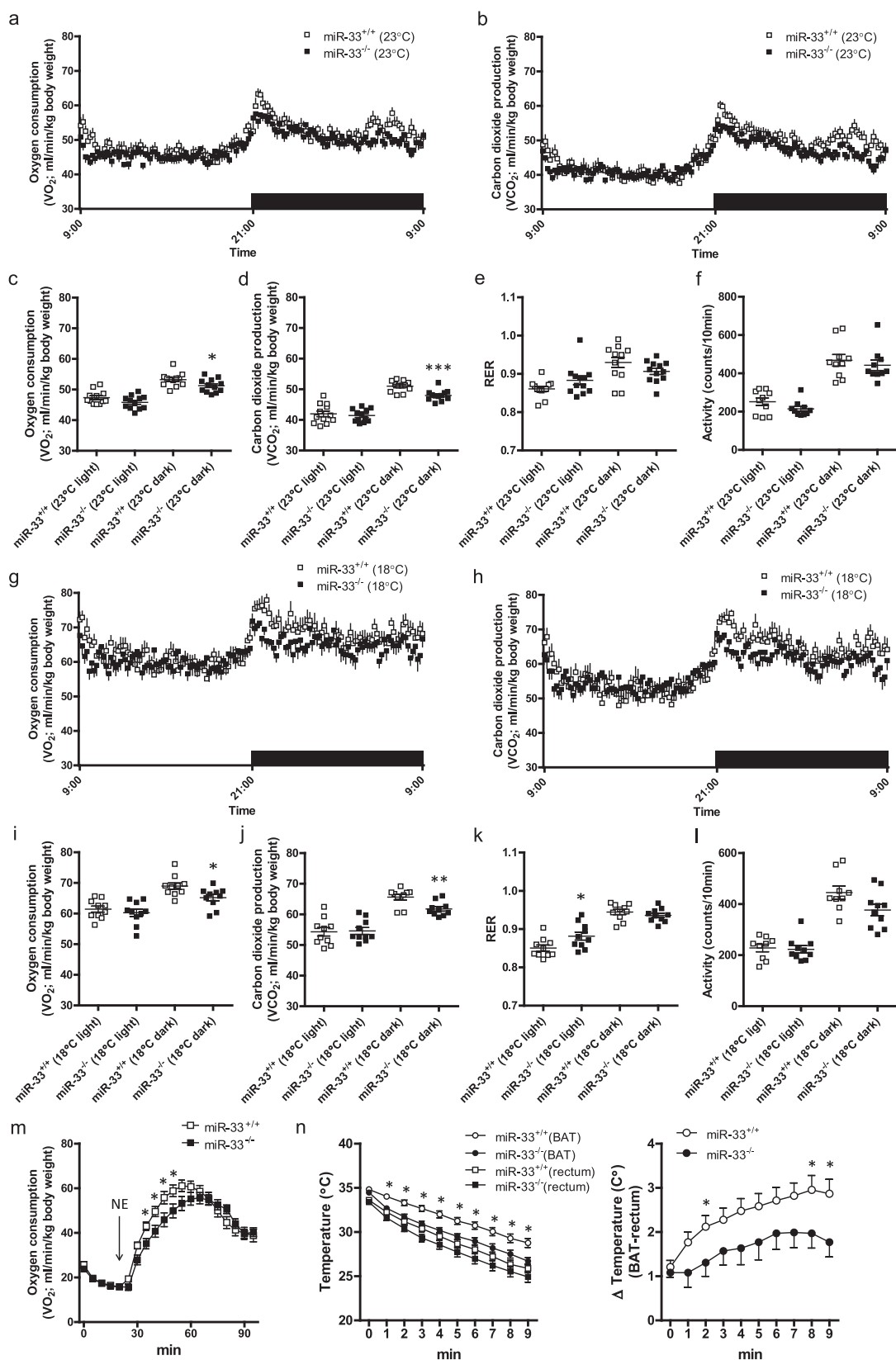

sympathetic premotor neurons that control BAT thermogenesis by directly innervating sympathetic preganglionic neurons at the spinal IML[29]. We found that the number of c-fos-positive cells in the rRPa was significantly reduced in miR-33$^{-/-}$ mice after 4 °C exposure for 2 h (Fig. 3d), indicating that the activation of

sympathetic outflow after cold exposure was decreased in miR-33$^{-/-}$ mice. Recent evidence has shown the importance of beige adipose tissue, which differentiates from WAT and serves similar functions as BAT. Differentiation of beige adipocytes is also under the control of sympathetic nerves[27]. We kept mice at 16 °C

**Fig. 2 miR-33$^{-/-}$ mice exhibit reduced oxygen consumption in a cold environment. a** Oxygen consumption rate in miR-33$^{+/+}$ and miR-33$^{-/-}$ mice kept at 23 °C. $n = 12$ mice per group, **$p < 0.01$, fdANOVA. **b** Carbon dioxide production rate in miR-33$^{+/+}$ and miR-33$^{-/-}$ mice kept at 23 °C. $n = 12$ mice per group, *$p < 0.05$, fdANOVA. **c** Mean oxygen consumption rate during the light and dark phase in miR-33$^{+/+}$ and miR-33$^{-/-}$ mice kept at 23 °C. $n = 12$ mice per group, *$p < 0.05$, two-sided unpaired $t$ test. **d** Mean carbon dioxide production rate in miR-33$^{+/+}$ and miR-33$^{-/-}$ mice kept at 23 °C. $n = 12$ mice per group. ***$p < 0.001$, two-sided unpaired $t$ test. **e** Respiratory exchange rate in miR-33$^{+/+}$ and miR-33$^{-/-}$ mice kept at 23 °C. $n = 12$ mice per group. **f** Locomotor activity of miR-33$^{+/+}$ and miR-33$^{-/-}$ mice kept at 23 °C. $n = 10$ mice per group. **g** Oxygen consumption rate in miR-33$^{+/+}$ and miR-33$^{-/-}$ mice at 2 days after transfer from 23 to 18 °C. $n = 10$ mice per group, $p < 0.05$, fdANOVA. **h** Carbon dioxide production rate in miR-33$^{+/+}$ and miR-33$^{-/-}$ mice at 2 days after transferring from 23 to 18 °C. $n = 10$ mice per group, $p < 0.05$, fdANOVA. **i** Mean oxygen consumption rate in miR-33$^{+/+}$ and miR-33$^{-/-}$ mice at 2 days after transferring from 23 to 18 °C. $n = 10$ mice per group, *$p < 0.05$, two-sided unpaired $t$ test. **j** Mean carbon dioxide production rate in miR-33$^{+/+}$ and miR-33$^{-/-}$ mice at 2 days after transferring from 23 to 18 °C. $n = 10$ mice per group. **$p < 0.01$, two-sided Mann–Whitney test. **k** Respiratory exchange rate in miR-33$^{+/+}$ and miR-33$^{-/-}$ mice at 2 days after transferring from 23 to 18 °C. $n = 10$ mice per group. *$p < 0.05$, two-sided unpaired $t$ test. **l** Locomotor activity of miR-33$^{+/+}$ and miR-33$^{-/-}$ mice at 2 days after transferring from 23 to 18 °C. $n = 9$ mice per group. **m** Maximum oxygen consumption after 1.0 mg/kg subcutaneous norepinephrine (NE) injection. miR-33$^{+/+}$ and miR-33$^{-/-}$ mice at 3 days after transfer from 23 to 18 °C were used. Measurements were performed at 30 °C chambers. $n = 15$, 13 mice per group, *$p < 0.05$, two-sided Mann–Whitney test for each time. **n** Serial measurement of BAT and rectal temperature in miR-33$^{+/+}$ and miR-33$^{-/-}$ mice at a cold condition. $n = 13$, 11 mice per group. *$p < 0.05$, two-sided Mann–Whitney test. All data are presented as mean ± SEM.

for 1 week, then analyzed their inguinal WAT. Induction of beige adipocyte markers including *Ucp1* was attenuated in miR-33$^{-/-}$ mice at both the mRNA and protein levels without TH modification (Fig. 3e–g). Taken together, these results show that miR-33$^{-/-}$ mice exhibited reduced sympathetic nerve activity to adipose tissues.

**Cold exposure induces hypothalamic miR-33, possibly via ER stress.** To clarify the relationship between sympathetic nerve activity and miR-33 in a cold environment, we measured the expression of miR-33 in the hypothalamus after cold exposure in wild-type mice. We found that hypothalamic miR-33 expression was elevated by intermittent cold exposure over 7 days, and this was accompanied by induction of its host gene, *Srebf2* (Supplementary Fig. 6a). *Srebf1* was also induced in this condition. Previous in vitro studies have shown that hypothermic conditions induce mild endoplasmic reticulum (ER) stress in cortical neurons[30], and it is known that ER stress induces miR-33 expression in macrophages[31] and astrocytes[32]. In our study, expression levels of ER stress markers *Bip* and *Chop* in the hypothalamus were mildly induced by cold exposure for 6 h, and these inductions were reverted by subsequent exposure to room temperature for 18 h (Supplementary Fig. 6b). This BIP induction was also confirmed at the protein level by western blotting (Supplementary Fig. 6c, d). Cleaved-caspase 3, one of the markers of apoptosis, was not detectable in this condition. In an in vitro study, we treated Neuro2a cells with thapsigargin, an ER stress inducer, and found that *Srebf2* and miR-33 were induced along with *Bip* and *Chop* even at a low concentration (Supplementary Fig. 6e). Significant inductions of BIP and CHOP protein levels were observed at a high concentration of thapsigargin (0.1 μM), which were accompanied by an increased level of cleaved-caspase 3 (Supplementary Fig. 6f, g). On the other hand, a low concentration of thapsigargin (0.01 μM) also induced BIP and CHOP but without an increase in cleaved-caspase 3 (Supplementary Fig. 6f, g). Because the miR-33 increase was prominent at a low concentration, mild ER stress may be important for miR-33 induction. HFD feeding, which is also known to induce ER stress in the hypothalamus[33], also induced *Srebf2* and miR-33 in the hypothalamus (Supplementary Fig. 2h). These results suggest that cold exposure and/or HFD feeding may induce miR-33 expression via increased ER stress in the hypothalamus.

**miR-33b$^{+/+}$ mice show increased BAT activity with higher sympathetic nerve activity.** Next, we analyzed humanized miR-33b$^{+/+}$ mice. Surprisingly, body core temperature was

significantly higher in miR-33b$^{+/+}$ mice than in littermate controls (miR-33b$^{-/-}$ mice), even at room temperature, in both genders (Fig. 4a, b). Lipid droplets were smaller and the cytoplasmic area was prominent by histological analysis (Fig. 4c and Supplementary Fig. 7a). We found that expression levels of *Ucp1* and *Ppargc1a* were significantly higher in the BAT of miR-33b$^{+/+}$ mice (Fig. 4d). The protein levels of UCP1 were also upregulated in the BAT of miR-33b$^{+/+}$ mice, while ABCA1 levels were suppressed (Fig. 4e, f). The levels of other thermogenic markers are shown in Supplementary Fig. 7b. Sympathetic nerve innervation to the BAT was comparable as evaluated by western blotting and immunohistochemistry for TH (Fig. 4e–g). In accordance with the higher expression of thermogenic genes, the oxygen consumption rate at room temperature was significantly higher in miR-33b$^{+/+}$ mice (Fig. 4h, i). Carbon dioxide production was significantly higher in miR-33b$^{+/+}$ mice in the light phase (Supplementary Fig. 7c). The RER and locomotor activity were similar between miR-33b$^{-/-}$ and miR-33b$^{+/+}$ mice (Supplementary Fig. 7d, e). The oxygen consumption rate was similar at 30 °C, whereas the difference was much more evident at 18 °C (Supplementary Fig. 7f–i). NE-induced maximum oxygen consumption at room temperature was significantly higher in miR-33b$^{+/+}$ mice (Fig. 4j). These results point to the higher activity of BAT thermogenesis in miR-33b$^{+/+}$ mice. NETO in the BAT was higher in miR-33b$^{+/+}$ mice (Fig. 4k). At room temperature, c-fos-positive cells in the rRPa were rare in miR-33b$^{-/-}$ control mice, whereas they were significantly increased in the brain of miR-33b$^{+/+}$ mice (Fig. 4l, m). In the cold condition, miR-33b$^{+/+}$ mice ameliorated body temperature decline and showed increased expression of thermogenic genes including *Ucp1*, *Ppargc1a*, *and Pparg* in the BAT. (Supplementary Fig. 7j, k). Moreover, blood pressure was slightly but significantly higher in miR-33b$^{+/+}$ mice (Supplementary Fig. 7l). Thus we infer that miR-33b$^{+/+}$ mice exhibit increased BAT activity via high sympathetic nerve activity.

**miR-33$^{f/f}$ DBH-Cre mice exhibit a phenotype similar to miR-33$^{-/-}$ mice.** Because sympathetic nerve activity was altered in miR-33$^{-/-}$ and miR-33b$^{+/+}$ mice, we crossed miR-33$^{f/f}$ mice with *DBH*-Cre mice, which express Cre recombinase under the control of the human dopamine β-hydroxylase promotor. In *DBH*-Cre mice, Cre recombinase is expressed in catecholamine-producing cells like noradrenergic and adrenergic cells, and the expression patterns have been previously reported in detail, including several hypothalamic nuclei such as the DMH[34]. We confirmed that the expression of Cre recombinase in *Dbh*-positive cells neither affected body temperature at both room temperature and in a cold environment (Supplementary Fig. 8a), nor altered

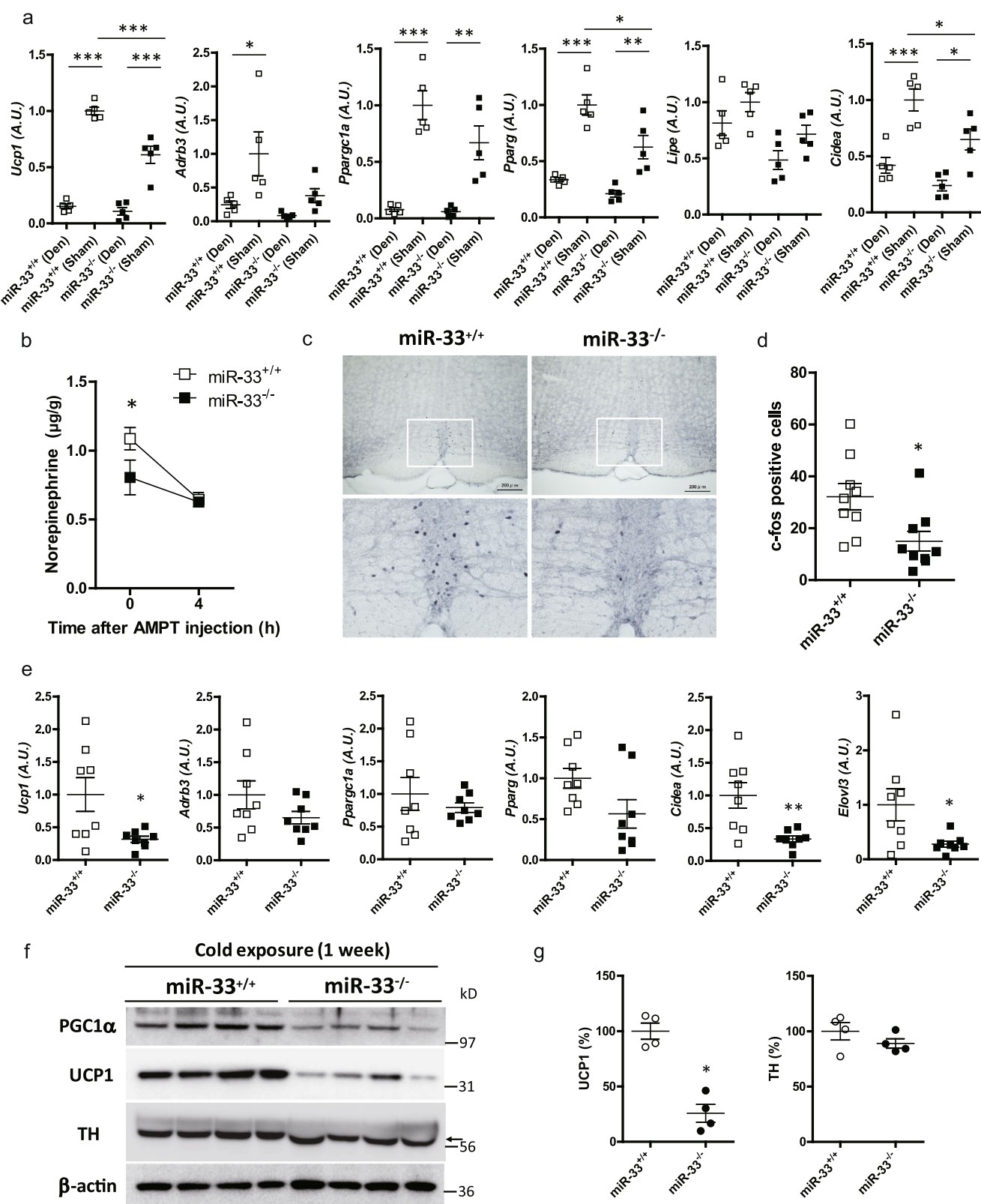

*Ucp1* expression in the BAT in a cold environment (Supplementary Fig. 8b). miR-33[f/f] *DBH*-Cre mice were born without apparent abnormality and showed normal growth until the age of 8 weeks (Fig. 5a). Successful recombination was confirmed by PCR analyses of genomic DNA from the brain and adrenal glands (Fig. 5b). The expression levels of miR-33 in the hypothalamus and adrenal glands were significantly reduced in miR-33[f/f] *DBH*-Cre mice

(Fig. 5c). At the age of 8 weeks, miR-33[f/f] *DBH*-Cre mice and their littermate control mice (miR-33[f/f] mice) were placed into a 4 °C environment for 5 h. Core body temperature decreased faster in miR-33[f/f] *DBH*-Cre mice than in the control mice (Fig. 5d). Histological analyses showed that many more lipid droplets remained in miR-33[f/f] *DBH*-Cre mice kept in a 4 °C environment for 5 h (Fig. 5e and Supplementary Fig. 8c). NETO in the BAT was found

**Fig. 3 miR-33$^{-/-}$ mice exhibit reduced sympathetic nerve activity. a** Quantitative PCR analysis of thermogenic genes in denervated or sham-operated BAT kept at 4 °C for 5 h. Left side-BAT was denervated and right side-BAT was sham-operated in miR-33$^{+/+}$ and miR-33$^{-/-}$ mice. At 1 week after the operation, mice were transferred to 4 °C for 5 h. $n = 5$ mice per group, ***$p < 0.001$, **$p < 0.01$, *$p < 0.05$, one-way ANOVA with Bonferroni's post hoc test. **b** Norepinephrine content in BAT of miR-33$^{+/+}$ and miR-33$^{-/-}$ mice before and after AMPT injection. $n = 5$–6 mice per group, *$p < 0.05$, two-way ANOVA with Bonferroni's post hoc test. **c** Representative images of c-fos immunohistochemistry in the rRPa of miR-33$^{+/+}$ and miR-33$^{-/-}$ mice kept at 4 °C for 2 h. $n = 9$ mice per group. Scale bar, 200 μm (upper). **d** Number of c-fos-positive cells in the rRPa of miR-33$^{+/+}$ and miR-33$^{-/-}$ mice kept at 4 °C for 2 h. $n = 9$ mice per group, *$p < 0.05$, two-sided unpaired $t$ test. **e** Quantitative real-time PCR analysis of beige adipocyte markers in inguinal WAT kept at 16 °C for 1 week in miR-33$^{+/+}$ and miR-33$^{-/-}$ mice. $n = 8$ mice per group, **$p < 0.01$, *$p < 0.05$, two-sided unpaired $t$ test. **f** Western blotting analysis of PGC1α, UCP1, TH, and β-actin in inguinal WAT at 16 °C for 1 week in miR-33$^{+/+}$ and miR-33$^{-/-}$ mice. $n = 4$ mice per group. $n = 4$ mice per group. The arrow indicates specific bands of TH staining. **g** Densitometry for UCP1 and TH in the inguinal WAT of miR-33$^{+/+}$ and miR-33$^{-/-}$ mice kept at 16 °C for 1 week. $n = 4$ mice per group, *$p < 0.05$, two-sided Mann–Whitney test. All data are presented as mean ± SEM.

to be reduced in miR-33$^{f/f}$ *DBH*-Cre mice (Fig. 5f). The expression levels of thermogenic genes, including *Ucp1*, *Adrb3*, and *Pparg* were downregulated in miR-33$^{f/f}$ *DBH*-Cre mice kept at 4 °C for 5 h (Fig. 5g).

Next, we assessed the metabolic parameters in miR-33$^{f/f}$ *DBH*-Cre mice. The oxygen consumption rate at 18 °C was significantly lower in miR-33$^{f/f}$ *DBH*-Cre mice in the dark phase (Fig. 6a, b). Carbon dioxide production, RER, and locomotor activity were similar between miR-33$^{f/f}$ and miR-33$^{f/f}$ *DBH*-Cre mice (Supplementary Fig. 8d–f). We evaluated the extent of browning of the inguinal WAT after exposure to 16 °C for 1 week, and found that the eosin-positive area was increased in miR-33$^{f/f}$ control mice after cold stimulation, but this increase was attenuated in miR-33$^{f/f}$ *DBH*-Cre mice (Fig. 6c). Beige adipocyte markers, such as *Ucp1*, *Cidea*, and *Elovl3*, were induced in miR-33$^{f/f}$ control mice by cold exposure; however, this induction was attenuated in miR-33$^{f/f}$ *DBH*-Cre mice (Fig. 6d). Protein levels of UCP1 were also attenuated in miR-33$^{f/f}$ *DBH*-Cre mice without TH change (Fig. 6e, f). To further characterize the metabolic phenotype of miR-33$^{f/f}$ *DBH*-Cre mice, we fed them an HFD. Unlike miR-33$^{-/-}$ mice, the food intake amounts were similar between miR-33$^{f/f}$ mice and miR-33$^{f/f}$ *DBH*-Cre mice, which indicates that loss of miR-33 in *Dbh*-positive neurons is not responsible for the increased food intake in miR-33$^{-/-}$ mice (Supplementary Figs. 8g and 1b). Bodyweight gain was more prominent in miR-33$^{f/f}$ *DBH*-Cre mice than miR-33$^{f/f}$ mice with HFD feeding (Supplementary Fig. 8h). These data indicate that loss of miR-33 in catecholamine neurons is responsible for impaired cold-induced thermogenesis and BAT activation, which contribute to the development of obesity.

**GABA$_A$ receptor subunit genes are altered by a miR-33 deficiency.** GABAergic neurons are involved in the central neural circuit controlling BAT thermogenesis[6,7]. Furthermore, a recent report showed that miR-33, which is induced in the hippocampus in fear conditioning, affects state-dependent memory by regulating several GABA-related genes[35]. To elucidate the molecular link between miR-33 and sympathetic nerve activity, candidate target genes of miR-33 were sought using public bioinformatics tools, and several GABA-related genes such as *Gabrb2*, *Gabra4*, and *Slc12a5* were found to be potential targets of miR-33 (TargetScan, http://www.targetscan.org). We measured the expression levels of *Abca1* and these GABA-related genes in the hypothalamus, medulla, and spinal cord of miR-33$^{+/+}$ and miR-33$^{-/-}$ mice. *Abca1*, one of the prominent target genes of miR-33, showed significantly higher expression levels in miR-33$^{-/-}$ mice (Fig. 7a). Likewise, the expression levels of *Gabrb2* and *Gabra4* were up-regulated in the hypothalamus of miR-33$^{-/-}$ mice (Fig. 7a). In contrast, the expression levels of these genes were down-regulated in the hypothalamus of miR-33b$^{+/+}$ mice (Supplementary Fig. 9a). Recently, we generated miR-33a, miR-33b, and miR-33a/b double knockout (KO) human iPS cells using CRISPR-Cas9-mediated genome editing[36]. These KO human iPS cells were differentiated into cortical

neurons using the quick embryoid body-like aggregate method (Supplementary Fig. 9b)[37]. Expression levels of miR-33a and miR-33b in these cells are shown in Fig. 7b. The expression levels of both *GABRB2* and *GABRA4* were significantly up-regulated in miR-33a/b double KO human iPS-derived neurons, while *SLC12A5* levels were not altered (Fig. 7c). Therefore, we focused on *Abca1*, *Gabrb2*, and *Gabra4* for further analysis. *Gabrb2* and *Gabra4* encode sub-units of the GABA$_A$ receptor, which is necessary for inhibitory GABAergic neurotransmission.

***Gabrb2* and *Gabra4* are responsible for altered thermogenesis in miR-33$^{-/-}$ mice.** To clarify whether up-regulated expressions of *Abca1*, *Gabrb2*, and *Gabra4* were responsible for the changes in thermogenesis we observed in miR-33$^{-/-}$ mice, we performed rescue experiments using an AAV vector. Although gene transfer to the brain is known to be difficult, intracerebroventricular (icv) injection of AAV within the first 24 h after birth yields widespread and persistent neuronal transduction, because it passes from the lateral ventricles through the immature ependymal barriers and into the brain parenchyma[38]. We chose AAV9 for this experiment because it has a high affinity for the central nervous system[39]. First, we infected AAV9-CMV-GFP into wild-type neonatal brains (P0) via icv injection, then analyzed brains at 8 weeks old. Fluorescence images of the whole brain and brain slices injected with the indicated particles of AAV9 are shown in Supplementary Fig. 10a, b. We found that neonatal icv injection of AAV9-GFP at a dose of 10$^{10}$ viral particles showed efficient gene transduction into the brain. To rescue the target genes of miR-33, we used a conditional short hairpin RNA (shRNA) AAV9 vector, by which shRNA is expressed only in cells that express Cre recombinase[40] (Supplementary Fig. 10c). Sequences of shRNAs for the in vivo rescue experiment were selected and validated in Neuro2a cells (Supplementary Fig. 10d). The conditional shRNA AAV9 vector was injected into the brains of neonates between male miR-33$^{f/f}$ *DBH*-Cre and female miR-33$^{f/f}$ mice, and then the mice were analyzed at 8 weeks old, as shown in the experimental scheme (Supplementary Fig. 10e, f). Each shRNA was expressed only in *Dbh*-positive cells of miR-33$^{f/f}$ *DBH*-Cre mice. Infection of the control and each conditional shRNA AAV9 in the brain did not affect the growth of mice, as shown by the comparable body weights at 8 weeks (Supplementary Fig. 10g).

When mice were infected with conditional control shRNA AAV9, body temperature decreased faster in miR-33$^{f/f}$ *DBH*-Cre mice than miR-33$^{f/f}$ mice at 4 °C, which reproduced the case without infection (Figs. 7d and 5d). Infection with conditional shRNA AAV9 against *Abca1* also did not affect the change in body temperature, indicating that *Abca1* is not involved in the phenotypic changes in miR-33$^{f/f}$ *DBH*-Cre mice (Supplementary Fig. 10h). In contrast, infection of conditional shRNA AAV9 against *Gabrb2* and *Gabra4* attenuated the decline in body temperature of miR-33$^{f/f}$ *DBH*-Cre mice at a level equal to or greater than that of miR-33$^{f/f}$ mice (Fig. 7e, f). Lipid droplets

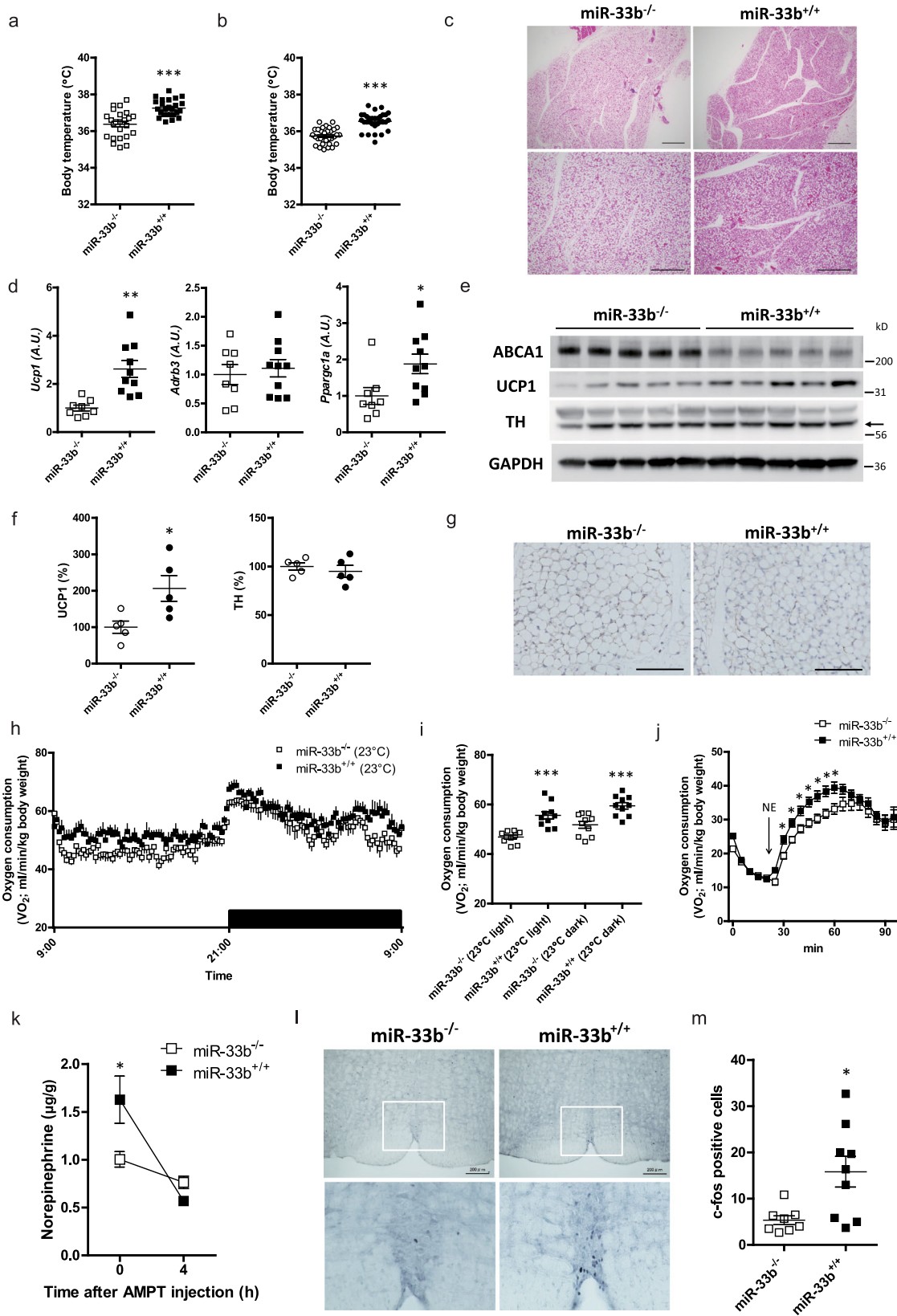

remained in the BAT of control conditional shRNA AAV9-infected miR-33[f/f] *DBH*-Cre mice after exposure at 4 °C for 5 h, whereas they were diminished in the BAT of conditional *Gabrb2* and *Gabra4* shRNA AAV9-infected miR-33[f/f] *DBH*-Cre mice (Fig. 7g and Supplementary Fig. 10i). In accordance with the histological changes, the gene expression of *Ucp1* in the BAT was significantly

suppressed in control conditional shRNA AAV9-infected miR-33[f/f] *DBH*-Cre mice, and this suppression was reversed in conditional *Gabrb2* and *Gabra4* shRNA AAV9-infected miR-33[f/f] *DBH*-Cre mice (Fig. 7h). The same tendency was observed in the expression of *Adrb3*. Taken together, up-regulation of *Gabrb2* and *Gabra4* in *Dbh*-positive neurons in the brain were responsible for the

**Fig. 4 miR-33b⁺/⁺ mice show increased BAT activity with higher sympathetic nerve activity. a** Core body temperature of miR-33b⁻/⁻ and miR-33b⁺/⁺ male mice kept at room temperature. $n = 24, 25$ mice per group, ***$p < 0.001$, two-sided unpaired $t$ test. **b** Core body temperature of miR-33b⁻/⁻ and miR-33b⁺/⁺ female mice kept at room temperature. $n = 32, 31$ mice per group, ***$p < 0.001$, two-sided unpaired $t$ test. **c** Representative images of HE staining for the BAT of miR-33b⁻/⁻ and miR-33b⁺/⁺ mice kept at room temperature. $n = 6$ mice per group. Scale bar, 100 μm (upper), 200 μm (lower). **d** Quantitative real-time PCR analysis of *Ucp1*, *Adrb3*, and *Ppargc1a* in the BAT of miR-33b⁻/⁻ and miR-33b⁺/⁺ mice kept at room temperature. $n = 8, 9$ mice per group, **$p < 0.01$, *$p < 0.05$, two-sided unpaired $t$ test. **e** Western blotting analysis of ABCA1, UCP1, TH, and GAPDH in the BAT of miR-33b⁻/⁻ and miR-33b⁺/⁺ mice kept at room temperature. $n = 5$ mice per group. The arrow indicates specific bands of TH staining. **f** Densitometry for UCP1 and TH in the BAT of miR-33b⁻/⁻ and miR-33b⁺/⁺ mice at room temperature. $n = 5$ mice per group, *$p < 0.05$, two-sided unpaired test. **g** Representative images of immunohistochemistry for TH in the BAT of miR-33b⁻/⁻ and miR-33b⁺/⁺ mice at room temperature. $n = 3$ mice per group. Scale bar, 100 μm. **h** Oxygen consumption rate of miR-33b⁻/⁻ and miR-33b⁺/⁺ mice kept at 23 °C. $n = 10$ mice per group, $p < 0.05$, fdANOVA. **i** Mean oxygen consumption rate during the light and dark phase in miR-33b⁻/⁻ and miR-33b⁺/⁺ mice kept at 23 °C. $n = 10$ mice per group, ***$p < 0.001$, two-sided unpaired $t$ test. **j** Maximum oxygen consumption by 1.0 mg/kg subcutaneous norepinephrine (NE) injection. miR-33b⁻/⁻ and miR-33b⁺/⁺ mice kept at room temperature were used. Measurement was performed at 30 °C in a chamber. $n = 15, 14$ mice per group, *$p < 0.05$, Mann–Whitney test for each time. **k** Norepinephrine contents in BAT of miR-33b⁻/⁻ and miR-33b⁺/⁺ mice before and after AMPT injection. $n = 6$–8 mice per group, *$p < 0.05$, two-way ANOVA with Bonferroni's post hoc test. **l** Representative images of c-fos immunohistochemistry of the rRPa of miR-33b⁻/⁻ and miR-33b⁺/⁺ mice kept at room temperature. $n = 8, 9$ mice per group. Scale bar, 200 μm (upper). **m** Number of c-fos positive cells at the rRPa of miR-33b⁻/⁻ and miR-33b⁺/⁺ mice kept at room temperature. $n = 8, 9$ mice per group, *$p < 0.05$, two-sided unpaired $t$ test. All data are presented as mean ± SEM.

---

phenotypic changes in miR-33⁻/⁻ mice. Thus, miR-33 promotes the activation of catecholamine neurons by decreasing the inhibitory effect of GABAergic inputs via down-regulation of *Gabrb2* and *Gabra4*, subunit-coding genes of GABA_A receptor.

## Discussion

In the current study, we showed that miR-33 in the brain plays an important role in the regulation of body temperature in cold environments by increasing sympathetic thermogenesis in BAT. We found that miR-33⁻/⁻ mice could not maintain their body temperature under cold conditions due to impaired BAT activation and a reduced metabolic rate. In these mice, sympathetic nerve activity was decreased, as indicated by a reduced NETO rate as well as a decreased number of c-fos-positive neurons in the rRPa of the medulla after cold exposure. miR-33^f/f *DBH*-Cre mice exhibited a similar phenotype to that of miR-33⁻/⁻ mice, indicating the importance of miR-33 in *Dbh*-positive cells. Our analysis of the miR-33⁻/⁻ hypothalamus and miR-33-deficient human iPS cell-derived neurons confirmed that *Abca1*, *Gabrb2*, and *Gabra4* were regulated by miR-33 in neurons. Rescue experiments using AAV9-mediated conditional shRNA demonstrated that upregulation of *Gabrb2* and *Gabra4* in *Dbh*-positive neurons in the brain was responsible for the phenotype of miR-33⁻/⁻ mice. In contrast, as a gain-of-function model of miR-33, miR-33b⁺/⁺ mice showed reduced levels of *Gabrb2* and *Garba4* in the hypothalamus and increased BAT activity with higher sympathetic nerve activity. Furthermore, we found that cold stress up-regulated miR-33 levels in the hypothalamus, possibly via ER stress. These results indicate the likely existence of a neural mechanism for controlling BAT thermogenesis involving hypothalamic miR-33, which exerts sympathetic nerve activation by suppressing GABAergic inhibitory neuronal transmission and can thereby regulate whole-body metabolism by modulating sympathetic nerve tone.

Adaptive thermogenesis is a fundamental mechanism that is indispensable for survival; therefore, it is tightly regulated by a central neural circuit. Glutamate-driven activation of BAT-sympathetic premotor neurons in the rRPa region of the medulla is essential for the thermoregulatory activation of BAT thermogenesis[6,7,41]. These neurons may be activated by several stimuli, including cold exposure, pyrogens such as prostaglandin E2, and psychological stress[29,42,43]. Sympathetic premotor neurons in the rRPa directly innervate sympathetic preganglionic neurons, which control the postganglionic sympathetic neurons innervating BAT. Thus, counting the number of c-fos-positive cells in the rRPa can allow for estimation of the activation of this pathway. We found that the number of c-fos-positive cells was

decreased in miR-33⁻/⁻ mice in a cold environment, while it was increased in miR-33b⁺/⁺ mice even at room temperature, in parallel with their BAT activity and metabolic rate. The activation of sympathetic premotor neurons in the rRPa is tightly regulated by the balance between glutamatergic excitatory and GABAergic inhibitory inputs from the upper neurons in the DMH and the POA of the hypothalamus, respectively[6,7,41]. Glutamatergic excitatory neurons from the DMH directly project to the pre-motor neurons at the rRPa to activate BAT thermogenesis, whereas GABAergic inhibitory neurons from the POA control the DMH and/or the rRPa to suppress BAT thermogenesis. It has been shown that nano injection of a GABA_A agonist into the DMH completely blocks BAT thermogenesis in response to cold exposure, pyrogens, or psychological stress[43–45]. Several GABA-related genes have been predicted to be the targets of miR-33[35]. Among these, we found that the expression of *Gabrb2* and *Gabra4*, which are subunits of GABA_A receptors, was elevated in the hypothalamus of miR-33⁻/⁻ brains and miR-33-deficient human iPS cell-derived neurons. In fact, the reduction of *Gabrb2* and *Gabra4* expression in *Dbh*-positive neurons rescued the phenotype of miR-33^f/f *DBH*-Cre mice. Therefore, miR-33 in some populations of catecholamine-producing neurons can enhance sympathetic nerve activity by targeting *Gabrb2* and *Gabra4* and thus suppressing upper thermoregulatory GABAergic inhibitory neurotransmission. Moreover, shivering thermogenesis, which was not examined in this study, is also important for thermogenesis in cold environments. Because regulation of the central nervous system for shivering thermogenesis is known to function similarly as in BAT thermogenesis[6], this might be also changed by miR-33 alternation in the brain to affect the maintenance of body temperature and whole-body metabolism.

Previous studies have shown that fear conditioning up-regulates miR-33 levels in the hippocampus[35], and miR-33 levels are also elevated in the hippocampus of behavioral stress-responsive mice[32]. Therefore, we speculated that cold stress might also increase miR-33 expression in the hypothalamus. Our results show that cold stress and HFD feeding augmented miR-33 expression in the hypothalamus, along with its host gene, *Srebf2*. Hypothermia has been reported to induce ER stress in cortical neurons[30] as occurs in the brain under HFD feeding[33]. ER stress is known to activate SREBP2[46], and transcription of *SRBEF2* is partly controlled by SREBP2 itself[47]. Since ER stress has also been shown to induce miR-33 in macrophages[31] and astrocytes[32], we focused on the putative role of ER stress for hypothalamic miR-33 induction. Our experiments with Neuro2a cells suggest that mild ER stress without apoptosis may be important for miR-33

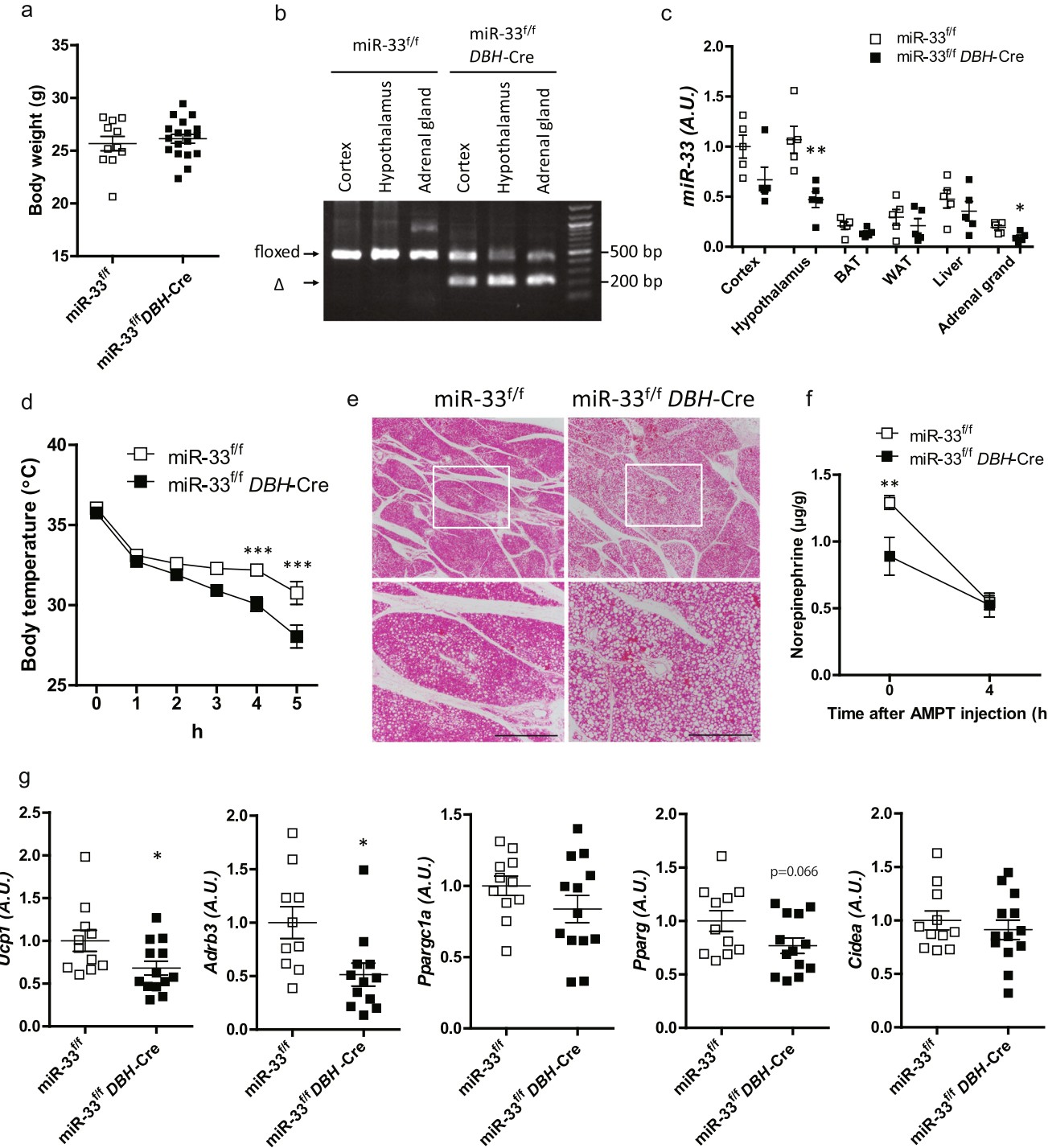

**Fig. 5 Generation and characteristics of miR-33$^{f/f}$ *DBH*-Cre mice. a** Body weight of miR-33$^{f/f}$ and miR-33$^{f/f}$ *DBH*-Cre mice at 8 weeks old. $n = 11$ or 18 mice per group. **b** PCR analysis of genome DNA from the brain and the adrenal gland of miR-33$^{f/f}$ and miR-33$^{f/f}$ *DBH*-Cre mice. A representative image from two independent experiments. **c** Quantitative real-time PCR analysis of miR-33 in organs of miR-33$^{f/f}$ and miR-33$^{f/f}$ *DBH*-Cre mice. $n = 5$ mice per group, \*\*$p < 0.01$, \*$p < 0.05$, two-sided Mann–Whitney test. The expression of miR-33$^{f/f}$ cortex is set to 1. **d** Serial core body temperature change of miR-33$^{f/f}$ and miR-33$^{f/f}$ *DBH*-Cre mice kept at 4 °C. $n = 11–13$ mice per group, \*\*\*$p < 0.001$, two-way repeated-measures ANOVA with Bonferroni's post hoc test. **e** Representative images of HE staining for the BAT of miR-33$^{f/f}$ and miR-33$^{f/f}$ *DBH*-Cre mice kept at 4 °C for 5 h. $n = 4$ mice per group. Scale bar, 200 μm (lower). **f** Norepinephrine contents in BAT before and after AMPT injection. $n = 5–7$ mice per group, \*\*$p < 0.01$, two-way ANOVA with Bonferroni's post hoc test. **g** Quantitative real-time PCR analysis of thermogenic genes in the BAT of miR-33$^{f/f}$ and miR-33$^{f/f}$ *DBH*-Cre mice kept at 4 °C for 5 h. $n = 10–13$ mice per group, \*\*$p < 0.01$, \*$p < 0.05$, two-sided unpaired $t$ test. All data are presented as mean ± SEM.

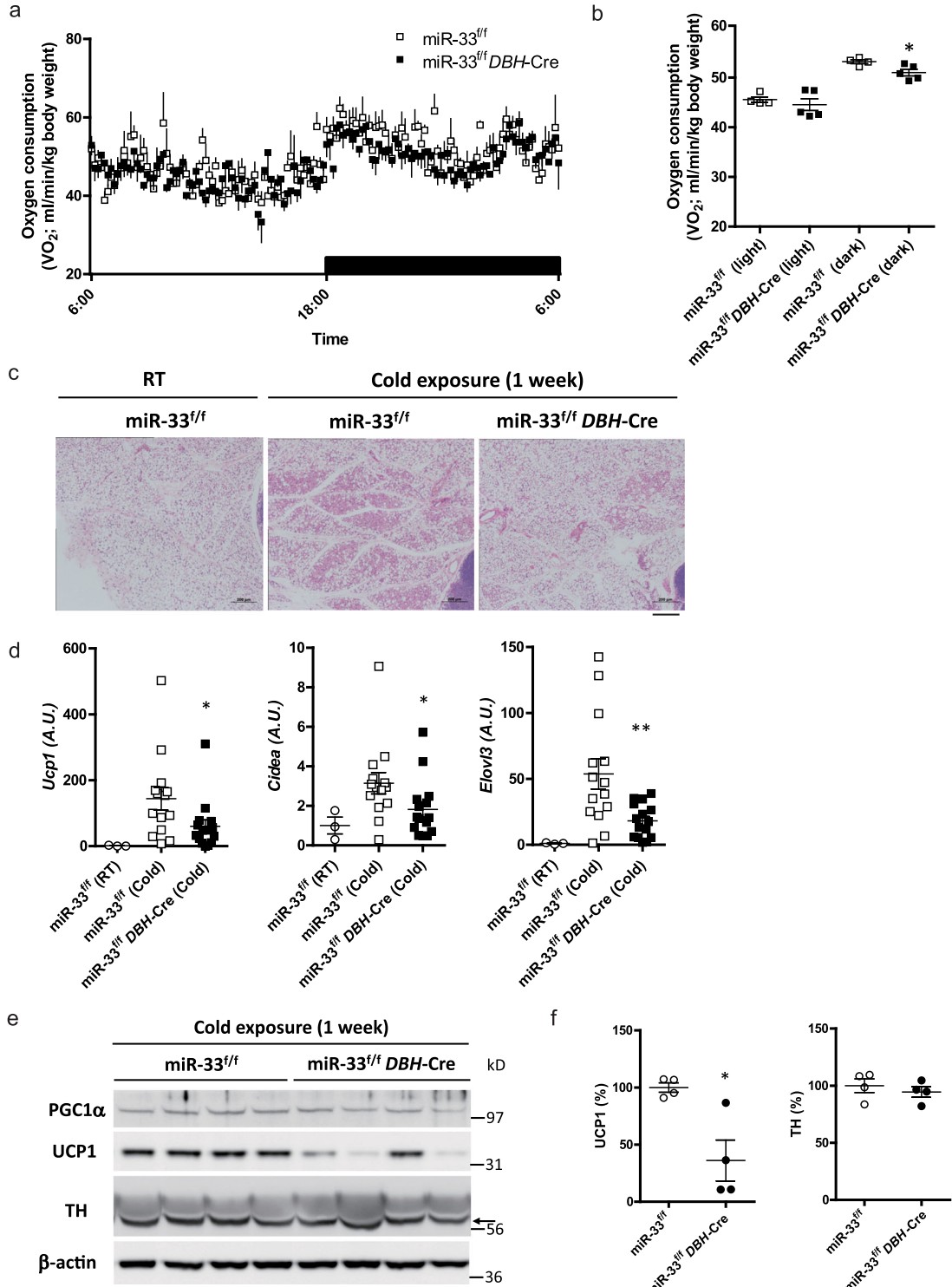

**Fig. 6 miR-33$^{f/f}$ *DBH*-Cre mice exhibit a phenotype similar to miR-33$^{-/-}$ mice. a** Oxygen consumption rate in miR-33$^{f/f}$ and miR-33$^{f/f}$ *DBH*-Cre mice kept at 18 °C $n = 4$ or 5 mice per group. **b** Mean oxygen consumption rate during the light and dark phase in miR-33$^{f/f}$ and miR-33$^{f/f}$ *DBH*-Cre mice kept at 18 °C. $n = 4$ or 5 mice per group. *$p < 0.05$, two-sided Mann–Whitney test. **c** Representative images of HE staining for inguinal WAT kept at 16 °C for 1 week in miR-33$^{f/f}$ and miR-33$^{f/f}$ *DBH*-Cre mice. $n = 3, 5, 6$ mice per group. Hematoxylin-strongly stained area of inguinal lymph node was taken into the picture to evaluate almost the same location in inguinal WAT. Scale bar, 200 μm. **d** Quantitative real-time PCR analysis of beige adipocyte markers in inguinal WAT kept at 16 °C for 1 week in miR-33$^{f/f}$ and miR-33$^{f/f}$ *DBH*-Cre mice. $n = 3$ mice in miR-33$^{f/f}$ mice at room temperature, $n = 14, 15$ mice per group at 16 °C, **$p < 0.01$, *$p < 0.05$, two-sided Mann–Whitney test. **e** Western blotting analysis of PGC1α, UCP1, TH, and β-actin in inguinal WAT kept at 16 °C for 1 week in miR-33$^{f/f}$ and miR-33$^{f/f}$ *DBH*-Cre mice. $n = 4$ mice per group. The arrow indicates specific bands of TH staining. **f** Densitometry for UCP1 and TH in the inguinal WAT of miR-33$^{f/f}$ and miR-33$^{f/f}$ *DBH*-Cre mice at 16 °C for 1 week. $n = 4$ mice per group, *$p < 0.05$, two-sided Mann–Whitney test. All data are presented as mean ± SEM.

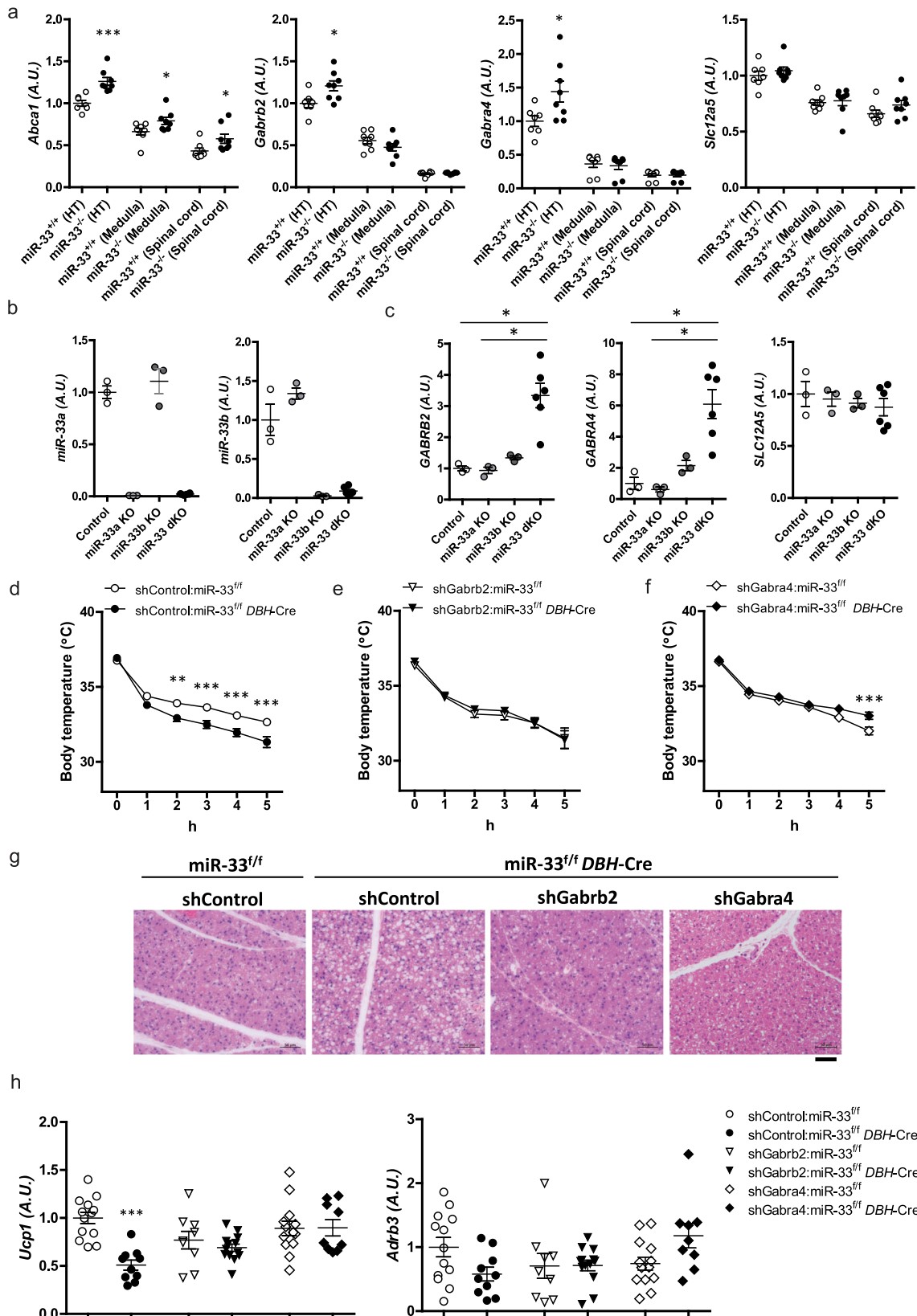

elevation. In the hypothalamus, ER stress markers were also mildly elevated in response to cold stress. The difference in the duration of miR-33 elevation between our in vitro and in vivo results may be due to the degree of ER stress, although further in vivo analysis is surely required. It takes several days for hypothalamic miR-33 to increase after cold exposure, which may

be an adaptive or complementary response to cold environments. Because miR-33 in the brain is induced by stresses such as fear conditioning, behavior stress, and cold stress, and given the ability of miR-33 to increase sympathetic nerve tone by suppressing GABAergic inhibitory transmission, this machinery is likely to be a physiological adaptive defense mechanism against

**Fig. 7 Gabrb2 and Gabra4 are the genes responsible for the phenotype of miR-33$^{-/-}$ mice. a** Quantitative real-time PCR analysis of *Abca1*, *Gabrb2*, *Gabra4*, and *Slc12a5* in the hypothalamus, medulla, and spinal cord of miR-33$^{+/+}$ and miR-33$^{-/-}$ mice. $n = 7$ or 8 mice per group ***$p < 0.001$, *$p < 0.05$, two-sided Mann–Whitney test. **b** Quantitative real-time PCR analysis of *miR-33a* and *miR-33b* in the control, miR-33a KO, miR-33b KO, and miR-33a/b double KO (dKO) human iPS cell-derived cortical neurons. $n = 3$ in control, miR-33a KO, miR-33b KO, $n = 6$ in miR-33 dKO. **c** Quantitative real-time PCR analysis of *GABRB2*, *GABRA4*, and *SLC12A5* in control, miR-33a KO, miR-33b KO, and miR-33a/b double KO iPS cell-derived cortical neurons. $n = 3$ in control, miR-33a KO, and miR-33b KO, $n = 6$ in miR-33 dKO, *$p < 0.05$, Kruskal–Wallis test with Dunn's post hoc test. **d** Serial core body temperature change of miR-33$^{f/f}$ and miR-33$^{f/f}$ DBH-Cre mice infected with conditional control shRNA AAV9 vector kept at 4 °C. $n = 13$, 10 mice per group, **$p < 0.01$, ***$p < 0.001$, two-way repeated-measures ANOVA with Bonferroni's post hoc test. **e** Serial core body temperature change in miR-33$^{f/f}$ and miR-33$^{f/f}$ DBH-Cre mice infected with conditional shRNA AAV9 vector against *Gabrb2* kept at 4 °C. $n = 9$, 13 mice per group. **f** Serial core body temperature change in miR-33$^{f/f}$ and miR-33$^{f/f}$ DBH-Cre mice infected with conditional shRNA AAV9 vector against *Gabra4* kept at 4 °C. $n = 13$, 9 mice per group, ***$p < 0.001$, two-way repeated-measures ANOVA with Bonferroni's post hoc test. **g** Representative images of HE staining for the BAT of the indicated mice kept at 4 °C for 5 h. $n = 4$, 4, 3, 4 mice per group. Scale bar, 50 μm. **h** Quantitative real-time PCR analysis of *Ucp1* and *Adrb3* in the BAT of control shRNA, *Gabrb2* shRNA, and *Gabra4* shRNA-infected mice kept at 4 °C for 5 h. The mean value of control shRNA-infected miR-33$^{f/f}$ mice was set to 1. $n = 13$, 10 mice per group for control shRNA, $n = 9$, 13 mice per group for *Gabrb2* shRNA, $n = 13$, 9 mice per group for *Gabra4* shRNA, ***$p < 0.001$, two-sided Mann–Whitney test. All data are presented as mean ± SEM.

life-threatening stress. However, the induction of miR-33 is relatively mild and takes time. In general, our results support that miR-33 is important for inducing sympathetic nerve activation, and suggest that its induction may be a complement to this neural mechanism under cold exposure or HFD feeding.

miR-33 is one of the best-characterized miRNAs[48,49]. Several previous studies, including our own, indicate that miR-33 regulates metabolic pathways such as cholesterol[10–12,18,19] and glucose homeostasis[50]. Because deletion or inhibition of miR-33 ameliorates atherosclerosis by improving lipid profiles and reducing inflammation, it shows promise as a therapeutic target for atherosclerosis-related diseases[13–15]. These beneficial effects originate from peripheral tissues or cells such as the liver and macrophages. Here, we found that miR-33 in the brain affects whole-body metabolism by modulating the sympathetic nerve tone. To the best of our knowledge, this is the first study to demonstrate that a certain miRNA in the central nervous system has substantial effects on metabolism by altering sympathetic nerve activity, and we identify crosstalk between miR-33, metabolic miRNA, and the central nervous system. Because increased sympathetic nerve activity is known to accelerate atherosclerosis by several mechanisms[51–54], its reduction via deletion or inhibition of miR-33 can contribute to the amelioration of atherosclerosis.

Rodents have only one miR-33 in an intron of the *Srebf2* gene, while humans have another, miR-33b, in an intron of the *SREBF1* gene. We previously generated miR-33b$^{+/+}$ mice, which have the human miR-33b sequence in the same intron of *Srebf1* as in humans[16], and we found the acceleration of atherosclerosis in these mice[17]. In contrast to miR-33$^{-/-}$ mice, these humanized mice exhibited reductions of *Gabrb2* and *Gabra4* in the hypothalamus, higher body temperature and metabolic rate, and higher sympathetic nerve activity even at room temperature. miR-33a and miR-33b have been shown to share the same genes as their targets by comprehensive transcriptome and bioinformatics analysis[55], therefore, the copy number of miR-33 is most likely to be important for these phenotypic changes. Because humans possess both miR-33a and miR-33b, as do miR-33b$^{+/+}$ mice, this neural circuit may have more impact in humans than in mice. In this study, *Srebf2* expression was induced by cold stress, HFD feeding, and thapsigargin treatment, while *Srebf1* was induced only by cold stress. Since there may be different pathways for inducing *Srebf1* expression other than ER stress[56], further analyses are needed.

Diet-induced thermogenesis is another form of adaptive thermogenesis that limits weight gain in response to caloric excess such as HFD feeding. HFD feeding for 2 weeks increased UCP1 expression in the BAT of miR-33$^{+/+}$ mice, but not in miR-33$^{-/-}$

mice. As discussed above, HFD feeding is known to induce ER stress in the hypothalamus[33]. We found that HFD feeding also up-regulated miR-33 in the hypothalamus. ER, stress-induced miR-33 can enhance sympathetic nerve activity by suppressing the GABA$_A$ receptor-mediated mechanism, leading to increased BAT thermogenesis. Thus, diet-induced thermogenesis probably shares the same pathway as cold-induced thermogenesis from the viewpoint of miR-33 and the GABA$_A$ receptors. A recent report showed that hunger signal stimulates GABAergic neurons in medullary reticular nuclei, which innervate the rRPa and inhibit BAT thermogenesis to reduce energy expenditure[57], which is compatible with and supports our findings. Accumulating evidence indicates that beige adipose tissue, which differentiates from WAT and functions similarly to BAT, is also important for whole-body metabolism and is expected to have clinical implications[27]. Because WAT is also under the control of the sympathetic nerve tone, we evaluated the extent of browning of WAT in this study. In accordance with the degree of sympathetic nerve activity, miR-33$^{-/-}$ mice and miR-33$^{f/f}$ DBH-Cre mice exhibited a reduced rate of browning of WAT under cold exposure. Previous reports have shown that miR-33 deficiency or long-term inhibition of miR-33 can lead to obesity with HFD feeding[20–22]. In this study, miR-33$^{-/-}$ mice showed reduced BAT activation, reduced metabolic rate, and susceptibility to HFD feeding, which were also observed in miR-33$^{f/f}$ DBH-Cre mice. Therefore, miR-33 deficiency or long-term inhibition of miR-33 may fail to induce HFD-induced thermogenesis and browning of WAT, which in part contributes to obesity with HFD feeding. On the other hand, food intake, another important factor in obesity, was increased in miR-33$^{-/-}$ mice but not in miR-33$^{f/f}$ DBH-Cre mice, which indicates that miR-33 in *Dbh*-positive neurons is not responsible for increased food intake in miR-33$^{-/-}$ mice. Further analysis is needed to elucidate this mechanism.

In conclusion, we identified a neural mechanism for the regulation of adaptive thermogenesis via miR-33. During cold stress, miR-33 in the brain (*Dbh*-positive neurons in particular) contributes to the maintenance of BAT thermogenesis and whole-body metabolism through increased sympathetic nerve activity by suppressing GABA$_A$ inhibitory neuronal transmission. This neural machinery may serve as an adaptive defense mechanism against cold stress and other stresses such as fear conditioning or behavior stress.

## Methods

**Animals**. miR-33$^{-/-}$, miR-33b$^{+/+}$, and miR-33$^{f/f}$ mice were originally generated in our laboratory as reported previously[12,16,58]. *DBH*-Cre mice were generated by Dr. Kazuto Kobayashi and provided by the RIKEN BRC through the National Bio-Resource Project of the MEXT, Japan[34]. To obtain miR-33$^{f/f}$ *DBH*-Cre mice, male *DBH*-Cre mice were crossed with female miR-33$^{f/f}$ mice. All of the in vivo

experiments were performed in C57BL/6J background mice, and littermates were used as controls. Mice were weaned at the age of 4 weeks and fed normal chow (NC) containing 4.5% fat until the age of 8 weeks, then used for experiments. In HFD experiments, 8-week-old mice were switched from NC to either 45% HFD (D12451; Research Diet, Inc. NJ) for 12 weeks, or 60% HFD (D12492) for 2 weeks. For cold exposure, mice were kept at 4 °C for the indicated times. All of the experimental protocols were approved by the Ethics Committee for Animal Experiments of Kyoto University. All animal experiments were performed in accordance with the ethical guidelines of the Animal Experimentation Committee of Kyoto University. Primers for genotyping and product size are listed in Supplementary Table 1.

**Cell culture and reagents.** Neuro2a cells were obtained from ATCC and maintained in Dulbecco's modified Eagle's medium (DMEM) supplemented with 10% FBS. Antibodies used in this study comprised: anti-ABCA1 antibody (NB400-105; Novus Biologicals, CO), anti-UCP1 antibody (U6382), anti-β-actin antibody (A5441) (Sigma-Aldrich, MO), anti-TH antibody (AB152), anti-goat-IgG biotin-conjugated antibody (AP180B), anti-rabbit-IgG biotin-conjugated antibody (AP182B) (Millipore, MA), anti-c-fos antibody (sc-52), anti-PGC1α antibody (H-300), anti-CHOP antibody (sc-575) (Santa Cruz Biotechnology, CA), anti-GAPDH antibody (#2118), anti-BIP antibody (#3183), anti-cleaved-caspase 3 antibody (#9664) (Cell Signaling Technology, MA), and anti-rabbit, anti-mouse, and anti-goat IgG HRP-linked antibodies (GE Healthcare, Inc., IL). NE, AMPT methyl ester hydrochloride, and thapsigargin were purchased from Sigma-Aldrich.

**Quantitative real-time PCR for mRNAs and miRNAs.** Total RNA was isolated using TriPure reagent (Sigma-Aldrich) from organs or cells. For coding genes, cDNAs were synthesized using a Verso cDNA synthesis kit (Thermo Fisher Scientific, MA), and quantitative real-time PCR (qRT-PCR) measurements used THUNDERBIRD SYBR qPCR Mix (TOYOBO, Japan) in accordance with the manufacturers' instructions. Expression levels were normalized with β-actin expression levels. miR-33a and miR-33b were measured using TaqMan MicroRNA Assays (Applied Biosystems, Inc., CA) in accordance with the manufacturer's instructions. Expression levels were normalized using U6 snRNA expression. Human Total RNA Master Panel II (Clontech, CA) and human adipose tissue RNA (Biochain, CA) were used for the distribution of genes in human organs. All analyses were performed using a StepOnePlus real-time PCR system and StepOne Software v2.3 (Applied Biosystems, Inc.). Primers used for quantitative real-time PCR in this study are listed in Supplementary Table 2.

**Western blotting.** Western blotting was performed using standard procedures. A total of 20 μg of protein was fractionated using NuPAGE 4–12% Bis–Tris gels (Thermo Fisher Scientific, MA) and transferred to a Protran nitrocellulose transfer membrane (GE Healthcare, Inc). The membrane was blocked using 1× phosphate-buffered saline (PBS) containing 5% non-fat milk for 1 h and incubated with a primary antibody (anti-ABCA1; 1:1000, anti-UCP1; 1:2000, anti-TH; 1:1000, anti-β-actin; 1:3000, anti-PGC1α; 1:500, anti-GAPDH; 1:3000, anti-BIP; 1:1000, anti-CHOP; 1:500, anti-cleaved-caspase 3; 1:500) overnight at 4 °C. After a washing step in PBS containing 0.05% Tween 20 (0.05% T-PBS), the membrane was incubated with the secondary antibody (anti-rabbit IgG HRP-linked; 1:2000, anti-mouse IgG HRP-linked; 1:2000) for 1 h at 4 °C. The membrane was then washed in 0.05% T-PBS and detected with ECL Western Blotting Detection Reagent (GE Healthcare), using an ImageQuant LAS4000 mini system (GE Healthcare, Inc, IL). Densitometry was performed using ImageJ software 1.44p.

**Measurement of rectal core body temperature.** The rectal core body temperature of mice was measured with a thermometer and probe for a mouse (AD-1687-M) (A & D company, Japan) at 10 A.M. For serial measurement of body temperatures, mice fasted for 2 h, then measurements were taken at 12 P.M., under a room temperature of 23 °C.

**Measurement of BAT and rectal temperature.** Anesthetized mice were placed on ice, and BAT and rectal temperature were recorded simultaneously every minute. A needle-type probe and thermometer were used for direct measurement of BAT temperature (AD-1220-100 and AD5601A) (A & D company, Japan).

**Assessment of the metabolic rate.** Male mice at 8 weeks old were kept in separate chambers at the indicated temperature for 3 days, and oxygen consumption rates were measured using an Oxymax indirect calorimetric system and analyzed by CLAX software v2.2.10 (Columbus Instruments, OH)[59]. For the experiment of miR-33^f/f *DBH*-Cre mice, male mice at 16 weeks of age were used.

**Assessment of maximum oxygen consumption rate.** For the assessment of the maximum oxygen consumption rate, male mice kept at the indicated temperature for 3 days were anesthetized with pentobarbital, and then transferred to a chamber at 30 °C. After measurement of basal rates for 30 min, mice were subcutaneously injected with NE (1 mg/kg), and oxygen consumption rate was measured for another 60 min[24,25].

**Measurement of FAO activity.** We measured FAO activity in the BAT using a non-radioactive enzymatic assay kit according to the manufacturer's instruction (BMR, NY)[60]. The measurement is based on the oxidation of octanoyl-CoA, which is coupled to NADH-dependent reduction of INT to INT-formazan. The formazan production from the same amount of BAT homogenates with or without octanoyl-CoA was measured by an ARVO X3 microplate reader at O.D. 492 nm (Perkin-Elmer, MA). The subtracted O.D. values presented here are proportional to the FAO activity.

**Measurement of NETO.** NETO was measured using AMPT as follows[25,28]. In brief, mice were divided into two groups and then BAT was collected without injection of AMPT (0 h) in one group. In the other group, BAT was collected 4 h after AMPT intraperitoneal injection (300 mg/kg). Mice were euthanized by decapitation without anesthesia in order to prevent NE release from sympathetic terminals. BAT was homogenized, and then NE concentrations were measured for the pre- and post-AMPT treatment groups. Values were calculated as NE content per gram of BAT.

**Surgical denervation of iBAT nerves.** Surgical denervation of nerves that innervate iBAT was performed as follows[28]. Under anesthesia with isoflurane, a midline incision in the skin along the upper dorsal surface was made. The medial, ventral surface of iBAT was exposed to visualize nerves beneath the pad. There are five intercostal nerves that unilaterally innervate each iBAT lobe and these nerves appear in bundles of two to three. All of these nerves were carefully cut using micro scissors with the use of a microscope so as not to cut the large Sulzer's vein. The left side of BAT was denervated and the right side of BAT was sham-operated. Mice were analyzed 1 week after the operation. Successful denervation was confirmed by NE content in BAT.

**Immunohistochemistry for c-fos-positive cells.** miR-33^−/− mice and their littermates were placed at 4 °C as cold exposure for 2 h. miR-33b^+/+ mice and their littermates were used at room temperature. After anesthetization, mice were perfused transcardially with ice-cold PBS (pH 7.4) followed by 4% PFA, then the brains were harvested. After overnight post-fixation with 4% PFA, brain samples were cryoprotected in a 30% sucrose solution. The brain tissues were embedded in O.C.T. Compound (Tissue-Tek, Japan) and sliced at a thickness of 40 μm. The slices were treated with methanol supplemented with 0.3% $H_2O_2$, blocked with 1% normal donkey serum containing 0.1% TritonX-100, and then incubated with a goat anti-c-fos primary antibody (1:200) overnight. After washing with T-PBS, slices were incubated biotin-conjugated anti-goat secondary antibody (1:500) for 1 h. After washing with T-PBS, sections were incubated with VECTASTAIN *Elite* ABC kit (Vector Laboratories, CA) for 30 min, then the color was developed using a DAB Substrate kit with nickel (Vector Laboratories). After washing with T-PBS, slices were embedded and coverslipped. c-fos-positive cells in the rRPa were counted. The mean number of c-fos-positive cells from six sequential sections are presented here. Data were obtained by Axio Observer 7 and Zen 2 pro software (Carl Zeiss, Inc).

**Intermittent cold exposure for mice.** To measure gene expression in the hypothalamus after intermittent cold exposure, wild-type mice were kept at 4 °C for 3 h and returned to room temperature for 21 h as one cycle. This cycle was repeated for 7 consecutive days.

**Measurement of blood pressure and heart rate.** Blood pressure and heart rate were measured using a noninvasive tail-cuff system (BP-98A, Softron). The mean of 8 measurements per mouse was used as values for blood pressure and heart rate.

**Generation of human iPS cell-derived cortical neurons.** miR-33a knockout, miR-33b knockout, and miR-33a/b double knockout human iPS cells were generated using CRISPR-Cas9-mediated genome editing as reported previously[36]. The human iPS cells used in this study were 201B7, which were generated by Dr. Shinya Yamanaka[61] from dermal fibroblasts (Cell Applications, Inc.) obtained from a healthy donor who gave informed consent. The iPS cell line is widely used for research and ethical approval is not required in Japan. These human iPS cells, including controls, were differentiated into cortical neurons using the quick embryoid body-like aggregate (SFEBq) method[37]. The use of human iPS cells was approved by the Ethics Committee of Kyoto University.

**Generation of AAV vector.** An AAV vector was generated using an AAV-2 Helper-Free System in accordance with the manufacturer's protocol (Cell Biolabs, Inc. CA). A plasmid with AAV capsid serotype 9 (pAAV-RC9) was obtained from Penn Vector Core at the University of Pennsylvania. A plasmid with AAV conditional shRNA (pAAV-dsRed-Sico-shRNA), in which the shRNA was expressed only when regulated by Cre recombinase was kindly provided by Dr. Marina Picciotto, Yale University[40]. The purification method was performed with slight modifications[62]. Briefly, AAV-293 cells (Agilent Technologies, CA) were transfected with pAAV-GFP or pAAV-conditional shRNA, pHelper, and pAAV-RC9 vector plasmids using a standard PEI (Polysciences, Inc. PA)-

mediated transfection method. 72 h after transfection, cells were collected and suspended in artificial CSF solution containing 124 mM NaCl, 3 mM KCl, 26 mM NaHCO$_3$, 2 mM CaCl$_2$, 1 mM MgSO$_4$, 1.25 mM KH$_2$PO$_4$ and 10 mM D-glucose. Following four freeze–thaw cycles, the cell lysates were treated with Benzonase nuclease (Millipore) at 45 °C for 15 min, then centrifuged twice at 16,000×$g$ for 10 min at 4 °C. The supernatant was used as the virus-containing solution. Quantitative real-time PCR was performed to measure the titer of the purified virus. Virus aliquots were then stored at -80 °C until used for the experiment.

**Screening and validation of shRNA sequence.** Two to three kinds of shRNA sequence against indicated genes were cloned into pAAV-conditional shRNA (pAAV-dsRed-Sico-shRNA). The sequences were confirmed by 3130xl Genetic Analyzer, Foundation Data Collection v3.1.1, and Sequence Analysis software v5.4 (Applied Biosystems, Inc). These plasmids are co-transfected with Cre-expression vectors into Neuro2a cells. 48 h later, the cells were collected and analyzed by quantitative real-time PCR. The most-effective sequence was selected for in vivo experiments. Sequences for shRNA used for the in vivo study are listed in Supplementary Table 3.

**Injection of AAV9 vector into neonatal mouse brain.** An AAV9 viral vector was injected into neonatal brains by icv injection as reported previously[38]. In brief, P0 neonatal mice were placed on a cold aluminum plate on ice to induce hypothermia anesthesia. After wiping the head of the pup with a cotton swab soaked in 70% ethanol, icv injection of AAV9 was performed using a 1701RN 33-gauge Neuros Syringe (Hamilton Company, NV). The needle was inserted at a site two-fifths of the distance from the lambda suture to each eye until a depth of 3 mm. Then, 5 × 10$^9$ viral particles per hemisphere were injected. The same procedure was performed with the contralateral ventricle. After completing viral injections into both hemispheres, the pup was placed back on a warming pad. AAV9 vector-injected mice were analyzed at 8 weeks of age.

**Statistics.** Data are presented as the mean ± standard error of means. Statistical comparisons were performed using either two-sided paired or unpaired $t$ test or Mann–Whitney test for the comparison of two groups, and either one-way or two-way or two-way repeated-measures analysis of variance with Bonferroni's post hoc test or Kruskal–Wallis test with post hoc Dunn's test for the multiple groups, based on the normality test, with $p < 0.05$ taken to indicate significance. fdANOVA was used for the analysis of the oxygen consumption curves. Exact $p$ values are provided in a Source Data file. Statistical and data analyses were performed using ImageJ 1.44p, GraphPad Prism 5.04 and 6.05 (GraphPad Software, Inc.), and R software 3.6.0.

**Reporting summary.** Further information on research design is available in the Nature Research Reporting Summary linked to this article.

## Data availability
The data that support the findings in this study are available from the corresponding authors upon reasonable request. Source data are provided with this paper.

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

## Acknowledgements

We would like to thank Dr. Kazuto Kobayashi and RIKEN BRC for providing DBH-Cre mice. This work was supported by the Ministry of Education, Culture, Sports, Science, and Technology (MEXT) and Japan Society for the Promotion of Science (JSPS) KAKENHI Grants 17K09860 and 20K08904 (T.H.), 1605297 (T.K.), 17H04177, 17H05599, and 20H03675 (K.O.). This work was also supported by AMED under grant JP19fk0210112 (K.O.), as well as grants from the Mochida Memorial Foundation for Medical and Pharmaceutical Research, Suzuken Memorial Foundation, Bristol Myers Squibb, Fujiwara Memorial Foundation (T.H.), and The Vehicle Racing Commemorative Foundation (K.O.).

## Author contributions

T.H. and K.O. designed the project; T.H., T. Nakao, Y.M., T. Nishino, S.M., F.N., T.W., N.S., A.O. and J.T. performed the experiments; T.H., T. Nakao, Y.M., Y.I., M.K., S.T., R.R.R., T.W., T.Y., S.X., C.O., S.M., K.M., M.N., Y.K., O.B., S.W., H.N., Y.N., T.S., T.K. and K.O. analyzed and interpreted the data; C.N., M.R.P., H.I., D.W., K.N. and T.S. contributed reagents, materials, and analysis tools; and T.H. and K.O. prepared the paper.

## Competing interests

The authors declare no competing interests.
