## [Peer Review File · Nature Communications]

Reviewers' comments, first round of review::

Reviewer #1 (Remarks to the Author):

The report from Horie et al. is interesting and could potentially contribute understanding of the regulation of thermogenesis BAT by miR-33 via sympathetic nerve activation. But the current version is premature. Specific instances are outlined in our responses to the authors:

1. It is somehow surprising that for all experiments where oxygen consumption is measured, the authors do NOT provide the rest of the metabolic cage data (RER, CO₂ production, activity, food intake). RER would be especially interesting to see how the lack of miR-33 impacts fuel preference and metabolic flexibility. Food intake and activity will also be important, especially since the authors have previously shown that these mice have increased food intake and a trend toward reduced activity.
2. The most interesting aspect of this manuscript is the characterization of the miR-33f/fDHB-cre mice, however this characterization is INCOMPLETE as no data on oxygen consumption, as well as other metabolic parameters, is provided under basal conditions or in response to cold stress.
3. It is also very misleading that the authors do not report any body weight data after HFD feeding or cold stress for young miR-33f/fDHB-cre mice. The presented data only focuses exclusively on the characterization of thermogenesis related phenotypes and does not directly assess any of the downstream metabolic implications, however the authors attempt to claim relevance for atherosclerosis and obesity, which seem very premature.
4. miR-33 expression in WAT and liver appears to be reduced (Supplementary Figure 7e). The authors should provide the statistics for this data and offer an explanation for this observation.
5. Mechanistic explanation is based largely on qRT-PCR data showing that "the expression levels of Gabrb2, Gabra4, and Slc12a5 were up-regulated in some regions of the brain of miR-33-/- mice". The data actually show a modest upregulation of Gabrb2 in the hypothalamus and Slc12a5 in the medulla and no significant changes in Gabra4. Similarly, all changes in miR-33b mice were non-significant. Moreover, this data does very little to demonstrate that these genes are upregulated in the specific neurons they claim are responsible for the phenotype, as these neurons make up only a fraction of the cells in these regions of the brain. This is highlighted by the fact that the expression of miR-33 (Supplemental figure 7e) is not significantly reduced in different brain regions and only shows a significant reduction of miR-33 in the adrenal gland of miR-33f/fDHB-cre mice.
6. The conclusion regarding ER stress is solely based on expression data and very superficial. For instance, there is no immunodetection of relevant ER stress modulators, more importantly no data on ER stress-induced cell death. This is key to support the conclusion.
7. Gene expression data in figure 6H-J should be reconstructed to include shControl, shGabrb2, and shGabra4 on the same graph with all values normalized to the control shRNA in f/f mice to demonstrate how constructs effect gene expression in control animals. Similarly, additional comparisons are needed for the body temperature experiments in Figure 6D-F.

Reviewer #2 (Remarks to the Author):

The manuscript by Horie and collaborators describes the role of miR-33 in BAT thermogenesis. From the study the authors concluded that miR-33 regulates adaptive thermogenesis via the sympathetic innervation. Several mouse models were used in the study included a new miR-33

floxed mouse model in which they selectively deleted miR-33 in DBH-expressing cells and a model with miR-33b overexpression.

This is an interesting study, however some concerns should be addressed:

While their conclusion that miR-33 regulates thermogenesis via the SNS is in part supported by some of the experiments, the conclusion that cold exposure-induces hypothalamic miR-33 via ER stress is weak as no experiments performed by the authors directly support this conclusion. They showed that while 6 hours cold exposure induced Bip and Chop gene expressions, it takes 7 days of cold exposure to induce miR33. Furthermore, while thapsigargin treatment induces Bip and Chop expression only at a concentration of 0.1uM, it takes a much lower dose of 0.01uM to induce miR-33.

In addition, the critical measurement of intrascapular BAT temperature is missing in the entire study in all animal models used. This is an important measurement especially when not always the markers of thermogenesis in the BAT are significantly different or when the difference between the experimental groups is marginal.

In all figures showing Western blots, graphs showing results from the densitometry analyses should be reported. This is true for Fig. 1e, 2f, 4e, 5g.

In Fig1i, the labeling of the X axis is wrong. miR33^{-/-}(23°C dark) should be miR33^{-/-}(18°C dark). In addition, the experiment in Fig1j is confusing. First NE administration (which is not described in the methods section) does not seem to increase oxygen consumption in the control if one compares the graph in Fig 1i with the graph in Fig 1j. Then, I would expect that NE administration would have a strong effect in the KO mice in which the activation of the SNS is lower. It would have been more informative to administer a dose of NE that was not effective in the controls hoping to show that the same dose had an effect in the KO.

According to Figure 4a and b, miR-33b^{+/+} mice showed significant increase in body temperature compared to miR-33b^{-/-} mice. However, in supplementary Figure 6g, body temperature in these same animal groups at time 0 is not significant. How do the authors reconcile this difference? In Figure 6b, miR-33a/b double knockout (KO) human iPS cells showed significant up-regulation of GABRB2 and GABRA4. However, single KOs showed no significant changes in GABRB2 and GABRA4 expressions suggesting that both miR-33a/b are required for gene regulation of GABRB2 and GABRA4. Since the mouse model generated overexpresses only miR-33b, how do the authors explain their results and the effect of the shRNA targeting Gabrb2 and Gabra4 (Figure 6g-j)? Could this be miR-33-independent? This should be considered especially because of the approach used to induce the downregulation of Gabrb2 and Gabra4 shRNA AAV9 infection of miR-33f/f DBH-cre mice.

Reviewer #3 (Remarks to the Author):

This study by Horie et al seeks to understand how miR-33, a well-known miRNA that has been reported to control metabolism and obesity, contributes to whole-body metabolism via its role in brown adipose tissue. Many previous studies have shown that loss of miR33 causes weight gain in the setting of high fat diet, and the authors seek in this study to better determine the cause of these observations. They use loss- and gain-of-function approaches in male mice, and use a Cre-mediated deletion of miR-33 using dopamine-β-hydroxylase Cre to delete miR33 in catecholamine-producing cells. They examine sympathetic innervation as a major mechanism controlling the observed defects in thermogenesis in miR-33^{-/-} mice, and show that miR33 controls the expression of GABAergic genes Gabra2 and Gabra4 in the hypothalamus.

This study is interesting and suggests novel mechanisms by which miR-33, which has previously been characterized mainly as a therapeutic target for cardiometabolic diseases, controls thermogenesis via activity in specific brain regions that control brown adipose tissue energy utilization. However, the conclusions in this study are not supported by the data, largely because the measurements were taken from different time points, across different exposures to cold temperatures, and many of the differences between groups are not quantified and appear small in effect size. While the use of the DBH-Cre mice are interesting, they confound the study because of their lack of specificity and the lack of characterization of the other cell(s) that may be influenced

by miR33 deletion.

Major comments:

The authors conclude that in the 2-week HFD study, there is "attenuation" of Ucp-1 expression with miR33^{-/-}. This is not well supported by the data shown in the paper. First, the UCP-1 IHC in Supp Fig 2 needs to be more clear and both the staining and the WB need to be quantified. In the image shown in the figure, there does not appear to be an appreciable reduction in Ucp1 protein. This is also true for many of the Western blots in the manuscript, which do not show quantification.

There is a discordance in the data regarding timing of the analyses. For example in Figure 1, some analyses are done after 6h of cold exposure, others after 48h. Please show protein expression after 6h, as this is purported to be when miR-33 is exerting physiologic effects (according to Fig 1a-c). Similarly, OCR was measured after 3 days at 18°C, yet gene expression, protein expression, BAT lipid droplets etc were not.

There are no measures of BAT fuel oxidation or electron transport chain activity, which is a primary mechanism that promotes thermogenesis and are downstream of Ucp1. The authors need to demonstrate that miR-33 impacts the uncoupling process in BAT.

The denervation experiments presented in Figure 2 are interesting and support that miR33 requires sympathetic nerve activity to stimulate thermogenic genes. However, given that BAT sympathetic activity is defective after denervation and unable to stimulate thermogenesis, it is important to show negative and positive control genes that are not controlled by denervation, as well as ones that are not under the control of miR-33 in order to conclude that these changes are due to loss of miR33.

Why do TH levels change in Figure 2f (WAT) but not in Figure 1e (BAT)? The authors do not discuss this yet the loss appears dramatic. These data confound whether miR-33 loss impairs denervation of the BAT or not.

The gene expression analyses in the hypothalamus in Figure 3 are done at different time points and after different patterns of cold exposure of the mice. Therefore, conclusions about miR33 levels and ER stress markers cannot be drawn, nor can mechanisms be extrapolated from the previous data in miR33^{-/-} mice as the cold exposure protocol was different and similar time points are not examined.

The data from miR33b^{+/+} mice are not consistent. First, there is a large difference in the number of animals analyzed in Figures 4 and Supplemental Fig 6. Second, there are few if any changes in gene expression of thermogenic genes at 4 degrees in miR33b mice compared to controls (only room temperature shows differences in expression).

According to the MGI, the DBH-Cre mouse model used in this system "Cre activity is detected in motor neurons of the cranial nerves and spinal cord, and sensory and enteric neurons". The authors therefore need to verify locomotor activity and other motor neuron features to ensure that miR-33 loss is not impacting whole-body motor function.

Similar to the above comment, according to the data in Supp Figure 4, skeletal muscle expresses a relatively high level of miR-33. Skeletal muscle also participates in thermogenesis and thus may be contributing to the role of miR33 in this regulation. In the studies of whole body metabolism of miR33^{-/-} mice, the authors should show physical activity measurements and characterize the skeletal muscle in response to cold exposure.

It is concerning that miR-33 levels were not significantly different in DBH-Cre mice according to Supp Figure 7e (except for in the adrenal gland). The authors justify the choice of the DBH-Cre system as this deletes floxed genes in "catecholamine-producing cells like noradrenergic and adrenergic cells [...] including several hypothalamic nuclei such as the DMH". There is a trend but no significant changes in miR33 expression in the hypothalamus. This region is also responsible for

food intake, which is not accounted for anywhere in this study, yet should be examined in both DBH-Cre and whole-body miR33^{-/-} mice.

Minor points:

In the short-term diet feeding study, why did they switch from 45% to using 60% HFD?

The axis on Figure 1a is compressed, and there are no error bars present. These should be shown.

Reviewer #4 (Remarks to the Author):

This is an excellent manuscript detailing the role of diminished BAT thermogenesis in the reduced cold defense and increased susceptibility to HFD in miRNA-33^{-/-} mice. The authors have significantly extended their study to demonstrate that these metabolic alterations arise from an increased GABAA-R expression in catecholamine neurons in the brain, resulting in a reduced sympathetic outflow to BAT. These results point to an important role for the miRNA-33 regulation of the GABAA-R expression in BAT regulating networks in the CNS. There is wide scientific interest in, and increasing appreciation of the roles of miRNAs in controlling the activity of a variety of tissues, and the high quality of this research will ensure that this report is highly cited.

1. There is no quantification for either the BAT staining data concerning lipid droplets or the gel data on protein levels. Since the authors describe significant differences in these measures, quantification and statistics should be provided.

2. Lines 128, 148, 164, 168 – the authors describe “trends” in their data, but this is not statistically valid: if there was no significant difference, then the data should be reported as “not different”, rather than the biased assessment of “trending” in the direction the authors would like.

3. As the authors are likely aware, “room temperature” (line 153) is actually a mild cold exposure for the mouse and should be described in this way. Exposure to 4C should be described as “deep cold”, keeping in mind that the threshold for cold nociceptors is around 15C.

4. Although they probably didn’t make any relevant observations, the authors might consider a brief mention in their Discussion of the potential for a similar regulation by miR-33 of shivering thermogenesis. The CNS pathways for shivering and for BAT thermogenesis are nearly overlapping (premotor neurons in rRPa excited by their antecedent neurons in the DMH), thus it is likely that miR-33 is influencing shivering in a manner similar to its regulation of BAT thermogenesis. Along these lines, shivering is a much more significant source of energy expenditure in humans than is BAT, thus it may be more relevant for weight loss in humans.

5. Lines 101-102 must be reworded to “In the current study, we analyzed several miR-33 genetically modified mice and found that miR-33 maintains BAT thermogenesis.”

6. Lines 204-205 (and elsewhere): NETO differences could arise for a variety of reasons, including differences in the ongoing level of sympathetic outflow, but also if there are different numbers of ganglion cells or different numbers of sympathetic axon terminals, the latter being an active research area as new growth factor molecules are being identified that are secreted by BAT and alter the density of its sympathetic innervation. Thus, the NETO data should be described as a potential surrogate for sympathetic nerve activity, but it is not a direct substitute for it.

7. Lines 233-234 (and elsewhere): although it is likely that ER stress induces miR-33, the authors have not demonstrated that the increased miR-33 with HFD or cold results from ER stress, but rather simply that both occur together. They would need to show that preventing ER stress during HFD or cold also prevents the induction of miR-33. Thus, this sentence (and similar conclusions elsewhere, including line 464) should be reworded to eliminate “indicate”, perhaps replacing with “suggest”.

8. Lines 284-286: can’t the authors make the more explicit conclusion that it is the miR-33 in some population of catecholamine neurons that provides the support of BAT thermogenesis? Further, in lines 348-350, can the authors explicitly state that their GABAA-R subunit genetic data supports the conclusion that miR-33 acts to decrease the level of GABAA-R in these catecholamine neurons?

9. Lines 392-396: The preceding part of this paragraph makes no mention of a potential catecholamine excitatory input to DMH, which could be a site of miR-33 action to support BAT

activation. The simple finding that Cre recombinase is found in the DMH of DBH-Cre mice does not logically indicate that "miR-33 in the DMH can enhance sympathetic nerve activity", unless there are catecholamine neurons in the DMH that project to rRPa. This concluding sentence should be rewritten.

Shaun F. Morrison

Response to Reviewer #1

We are grateful to reviewer #1 for the important comments and useful suggestions that have helped us to improve our manuscript considerably. As indicated in the responses that follow, we have taken all these comments and suggestions into account in the revised version of our manuscript.

The report from Horie et al. is interesting and could potentially contribute understanding of the regulation of thermogenesis BAT by miR-33 via sympathetic nerve activation. But the current version is premature.

Specific instances are outlined in our responses to the authors:

1. It is somehow surprising that for all experiments where oxygen consumption is measured, the authors do NOT provide the rest of the metabolic cage data (RER, CO₂ production, activity, food intake). RER would be especially interesting to see how the lack of miR-33 impacts fuel preference and metabolic flexibility. Food intake and activity will also be important, especially since the authors have previously shown that these mice have increased food intake and a trend toward reduced activity.

Thank you very much for these valuable comments. We collected data records from the metabolic cages and analyzed CO₂ production, RER, and locomotor activity in miR-33^{-/-} mice both at room temperature and in a cold environment. We found that VCO₂ was significantly reduced in miR-33^{-/-} mice in the dark phase both at room temperature and in a cold environment. On the other hand, RER and locomotor activity were not significantly different, although some tendency was observed. These data are shown in Fig. 2b, 2d-2f, 2h, and 2j-2l. Moreover, we checked the food intake of mice fed on an HFD and found that it was slightly, but still significantly, higher in miR-33^{-/-} mice, as in a previous report, which is shown in Supplementary Fig. 1b. We also analyzed CO₂ production, RER, and locomotor activity in miR-33b^{+/+} mice, which are presented in Supplementary Fig. 6e-6g. VCO₂ was significantly higher in miR-33b^{+/+} mice in the light phase. RER and locomotor activity were comparable between miR-33b^{-/-} and miR-33b^{+/+} mice.

We added and revised the description in the main text as follows.

•Lines 166–173 in the revised manuscript;

“On the other hand, the carbon dioxide production was significantly reduced in miR-33^{-/-} mice in the dark phase (Fig. 2b and 2d). The respiratory exchange rate (RER) and locomotor activity were similar between miR-33^{+/+} and miR-33^{-/-} mice (Fig. 2e and 2f). After 2 days of cold exposure at 18°C, the difference in the oxygen consumption became evident and significantly reduced in miR-33^{-/-} mice in the dark phase (Fig. 2g and 2i). The carbon dioxide production was also significantly reduced in miR-33^{-/-} mice (Fig. 2h and 2j). The RER and locomotor activity were not different between miR-33^{+/+} and miR-33^{-/-} mice in this condition (Fig. 2k and 2l).”

•Lines 119-120 in the revised manuscript;

“Food intake of an HFD was slightly but significantly higher in miR-33^{-/-} mice as reported previously (Supplementary Fig. 1b).”

•Lines 269-271 in the revised manuscript;

“The carbon dioxide production was significantly higher in miR-33b^{+/+} mice in the light phase (Supplementary Fig. 6e). The RER and locomotor activity were similar between miR-33b^{-/-} and miR-33b^{+/+} mice (Supplementary Fig. 6f and 6g).”

2. The most interesting aspect of this manuscript is the characterization of the miR-33f/fDBH-cre mice, however this characterization is INCOMPLETE as no data on oxygen consumption, as well as other metabolic parameters, is provided under basal conditions or in response to cold stress.

Thank you very much for these important comments. We analyzed miR-33^{ff} *DBH-Cre* mice and showed the metabolic cage data in a cold environment in Supplementary Fig. 7g-k. Due to availability, we used mice at the age of 16 weeks old in this experiment, whose body weights were approximately 5 g heavier than those at the aged 8 weeks old, but similar between the genotypes (miR-33^{ff}: 30.30±0.56g, miR-33^{ff} *DBH-Cre*: 31.04±0.63g). Thus, the data of VO₂ and VCO₂ were relatively lower than the others. However, VO₂ was significantly lower in miR-33^{ff} *DBH-Cre* mice in the dark phase, which was compatible with the data of miR-33^{-/-} mice. VCO₂, RER, and locomotor activity were similar between miR-33^{ff} and miR-33^{ff} *DBH-Cre* mice. We also checked the amounts of food intake, which are shown in Supplementary Fig. 7m. Food intake was similar between miR-33^{ff} and miR-33^{ff} *DBH-Cre* unlike miR-33^{-/-} mice. These data indicate that *DBH*-positive neurons are not responsible for the increased amounts of food intake in miR-33^{-/-} mice.

We added and revised the description in the main text as follows.

•Lines 304–308 in the revised manuscript;

“Next, we assessed the metabolic parameters in miR-33^{ff} *DBH*-Cre mice. The oxygen consumption rate at 18°C was significantly lower in miR-33^{ff} *DBH*-Cre mice in the dark phase, which was compatible with the data of miR-33^{-/-} mice (Supplementary Fig. 7g and 7h). The carbon dioxide production, RER and locomotor activity were similar between miR-33^{ff} and miR-33^{ff} *DBH*-Cre mice (Supplementary Fig. 7i-7k).”

•Lines 314–318 in the revised manuscript;

“To further characterize the metabolic phenotype of miR-33^{ff} *DBH*-Cre mice, we fed an HFD to these mice. Unlike miR-33^{-/-} mice, the amounts of food intake were similar between miR-33^{ff} mice and miR-33^{ff} *DBH*-Cre mice, which indicates that loss of miR-33 in *DBH*-positive neurons is not responsible for the increased food intake in miR-33^{-/-} mice (Supplementary Fig. 7m and 1b).”

•Lines 502–505 in the revised manuscript;

“On the other hand, food intake is another important factor for obesity. It was increased in miR-33^{-/-} mice but not in miR-33^{ff} *DBH*-Cre mice, which indicates that miR-33 in *DBH*-positive neurons is not responsible for the increased food intake in miR-33^{-/-} mice. Further analysis is needed to elucidate this mechanism.”

3. It is also very misleading that the authors do not report any body weight data after HFD feeding or cold stress for young miR-33ff/DHB-cre mice. The presented data only focuses exclusively on the characterization of thermogenesis related phenotypes and does not directly assess any of the downstream metabolic implications, however the authors attempt to claim relevance for atherosclerosis and obesity, which seem very premature.

Thank you very much for this important suggestion. We fully agree with this comment.

To assess the impact of miR-33 in *DBH*-positive neurons for obesity formation, we fed an HFD to miR-33^{ff} and miR-33^{ff} *DBH*-Cre mice. As shown in Supplementary Fig. 7n, miR-33^{ff} *DBH*-Cre mice gained much more weight than miR-33^{ff} mice, although the amount of food intake was not changed

(Supplementary Fig. 7m). These data indicate that loss of miR-33 in *DBH*-positive neurons contributes to obesity formation by decreasing energy expenditure via reduced BAT activation and browning of Inguinal WAT.

We added and revised the description in the main text as follows.

•Lines 314–322 in the revised manuscript;

“To further characterize the metabolic phenotype of miR-33^{ff} *DBH*-Cre mice, we fed an HFD to these mice. Unlike miR-33^{-/-} mice, the amounts of food intake were similar between miR-33^{ff} mice and miR-33^{ff} *DBH*-Cre mice, which indicates that loss of miR-33 in *DBH*-positive neurons is not responsible for the increased food intake in miR-33^{-/-} mice (Supplementary Fig. 7m and 1b). Body weight gain was more prominent in miR-33^{ff} *DBH*-Cre mice than miR-33^{ff} mice with HFD feeding (Supplementary Fig. 7n). These data indicate that loss of miR-33 in catecholamine neurons is responsible for the impaired cold-induced thermogenesis and BAT activation, which contribute to obesity formation.”

•Lines 498-500 in the revised manuscript;

“In this study, miR-33^{-/-} mice showed reduced BAT activation, reduced metabolic rate and a susceptibility to HFD feeding, which were also observed in miR-33^{ff} *DBH*-Cre mice.”

4. miR-33 expression in WAT and liver appears to be reduced (Supplementary Figure 7e). The authors should provide the statics for this data and offer an explanation for this observation.

Thank you very much for this valuable suggestion. Related to the next question and response, we increased the number of mice for analysis and re-analyzed the data in Supplementary Fig. 7e (Mann–Whitney test). As a result, the expression of miR-33 in the hypothalamus ($p=0.0079$) and adrenal gland ($p=0.0317$) were significantly reduced in miR-33^{ff} *DBH*-Cre mice. We also compared statistics for the WAT and the liver. However, the expression of miR-33 in WAT ($p=0.6905$) and the liver ($p=0.4206$) were not significantly reduced. Similarly, there was no statistically significant reduction in the cortex ($p=0.0556$) and BAT ($p=0.2222$). We included a new graph in Supplementary Fig. 7e.

We revised the description in the main text as follows.

•Lines 295–297 in the revised manuscript;

“The expression levels of miR-33 in the hypothalamus and adrenal glands were significantly reduced in miR-33^{ff} *DBH*-Cre mice (Supplementary Fig. 7e).”

5. Mechanistic explanation is based largely on qRT-PCR data showing that “the expression levels of *Gabrb2*, *Gabra4*, and *Slc12a5* were up-regulated in some regions of the brain of miR-33^{-/-} mice”. The data actually show a modest upregulation of *Gabrb2* in the hypothalamus and *Slc12a5* in the medulla and no significant changes in *Gabra4*. Similarly, all changes in miR-33b mice were non-significant. Moreover, this data does very little to demonstrate that these genes are upregulated in the specific neurons they claim are responsible for the phenotype, as these neurons make up only a fraction of the cells in these regions of the brain. This is highlighted by the fact that the expression of miR-33 (Supplemental figure 7e) is not significantly reduced in different brain regions and only shows a significant reduction of miR-33 in the adrenal gland of miR-33f/*DHB*-cre mice.

Thank you very much for your important comments. As described above, we increased the number of mice for analysis and found that the expression of miR-33 in the hypothalamus and adrenal glands were significantly reduced in miR-33^{ff} *DBH*-Cre mice, as shown in Supplementary Fig. 7e. We also increased the number of mice to analyze the expression of the potential target genes in the brain of miR-33^{+/+} and miR-33^{-/-} mice (from n=5–6 each to n=7–8 each). As shown in Fig. 7a, the expression of *Abca1* was significantly increased in all regions of the brain. The expression of *Gabra4* was also significantly up-regulated in the hypothalamus of miR-33^{-/-} mice as well as *Gabrb2*. However, the difference in *Slc12a5* in the medulla disappeared in this analysis. We revised new graphs in Fig. 7a. We further increased the number for analysis of these genes in the brain of miR-33b^{-/-} and miR-33b^{+/+} mice (from n=5 each to n=9 each). As a result, the difference of *Abca1*, *Gabrb2* and *Gabra4* reached to statistical significance, as shown in Supplementary Fig. 8a. These data indicate that the expression of miR-33 actually regulates the expression of *Abca1*, *Gabrb2*, and *Gabra4* in the hypothalamus.

We added and revised the description in the main text as follows.

•Lines 295–297 in the revised manuscript;

“The expression levels of miR-33 in the hypothalamus and adrenal glands were significantly reduced in miR-33^{ff} *DBH*-Cre mice (Supplementary Fig. 7e).”

•Lines 333–337 in the revised manuscript;

“*Abca1*, one of the prominent target genes of miR-33, showed significantly higher expression levels in miR-33^{-/-} mice (Fig. 7a). Likewise, the expression levels of *Gabrb2* and *Gabra4* were up-regulated in the hypothalamus of miR-33^{-/-} mice (Fig. 7a). In contrast, the expression levels of these genes were down-regulated in the hypothalamus of miR-33b^{+/+} mice (Supplementary Fig. 8a).”

6. The conclusion regarding ER stress is solely based on expression data and very superficial. For instance, there is no immunodetection of relevant ER stress modulators, more importantly no data on ER stress-induced cell death. This is key to support the conclusion.

Thank you very much for this important suggestion. We performed western blotting for BIP, CHOP, and cleaved-caspase 3 in Fig. 3 (Fig. 4 in the revised manuscript). BIP expression in the hypothalamus was significantly increased by cold exposure, whereas cleaved-caspase 3 and CHOP were not detectable, which suggested that ER stress by cold stress is mild. In the experiment of Neuro2a cells, a high concentration of thapsigargin (0.1 μM) induced the expression of BIP, CHOP along with cleaved-caspase 3. On the other hand, low concentration of thapsigargin (0.01 μM) also induced BIP and CHOP without an increase in cleaved-caspase 3. Because miR-33 elevation was more prominent in low concentration (0.01 μM) than high concentration (0.1 μM), mild ER stress without cell death may be enough and efficient for miR-33 induction. The data of immunoblotting are shown in Fig. 4c, 4d, 4f, and 4g.

We added and revised the description in the main text as follows.

•Lines 241–252 in the revised manuscript;

“The expression levels of ER stress markers *Bip* and *Chop* in the hypothalamus were mildly induced by cold exposure for 6 h, and these inductions were reverted by subsequent exposure to room temperature for 18 h (Fig. 4b). This BIP induction was also confirmed at the protein level by western blotting (Fig. 4c and 4d). Cleaved-caspase 3, one of the markers of apoptosis, was not detectable in this condition. In an *in vitro* study, we treated Neuro2a cells with

thapsigargin, an ER stress inducer, and found that *Srebf2* and miR-33 were induced along with *Bip* and *Chop* even at a low concentration (Fig. 4e). Significant inductions of BIP and CHOP protein levels were observed at a high concentration of thapsigargin (0.1 μ M), which were accompanied with an increased level of cleaved-caspase 3 (Fig. 4f and 4g). On the other hand, a low concentration of thapsigargin (0.01 μ M) also significantly induced BIP and CHOP but without an increase in cleaved-caspase 3 (Fig. 4f and 4g). Because the miR-33 increase was prominent at a low concentration, mild ER stress may be sufficient for miR-33 induction.”

·Lines 443–448 in the revised manuscript;

“The experiments with Neuro2a cells suggest that mild ER stress without apoptosis may be sufficient for miR-33 elevation. In the hypothalamus, ER stress markers were also mildly elevated in a cold stress condition. In fact, although a significant BIP induction was observed, an increase in cleaved-caspase 3 was not observed in this condition, which indicates that ER stress evoked by cold stress is mild. The difference in the duration for miR-33 elevation between *in vitro* and *in vivo* may be due to the degree of ER stress.”

7. Gene expression data in figure 6H-J should be reconstructed to include shControl, shGabrb2, and shGabra4 on the same graph with all values normalized to the control shRNA in f/f mice to demonstrate how constructs effect gene expression in control animals. Similarly, additional comparisons are needed for the body temperature experiments in Figure 6D-F.

Thank you very much for this meaningful advice. Accordingly, we reanalyzed the data and reconstructed the expression of *Ucp1* and *Adrb3* in shControl, shGabrb2 and shGabra4 on the same graph with all values normalized to the control shRNA in miR-33^{ff} mice, which are shown in Fig. 7h in the revised manuscript. In the case of body temperature, we tried to unify the data of Fig. 6d-f into one graph. However, the graph became very complicated and difficult to understand. Because real-time PCR data (*Ucp1* and *Adrb3*) are relative values, it is reasonable to reconstruct to one graph as above. However, body temperature data are absolute values and all of the three experiments were performed independently. Thus, we would like to show the three graphs

separately as an original version (Fig. 7d-f in the revised version).

Response to Reviewer #2

We are grateful to reviewer #2 for their important comments and useful suggestions that have helped us to improve our manuscript considerably. As indicated in the responses that follow, we have taken all these comments and suggestions into account in the revised version of our manuscript.

The manuscript by Horie and collaborators describes the role of miR-33 in BAT thermogenesis. From the study the authors concluded that miR-33 regulates adaptive thermogenesis via the sympathetic innervation. Several mouse models were used in the study included a new miR-33 floxed mouse model in which they selectively deleted miR-33 in DBH-expressing cells and a model with miR-33b overexpression.

This is an interesting study, however some concerns should be addressed:

1. While their conclusion that miR-33 regulates thermogenesis via the SNS is in part supported by some of the experiments, the conclusion that cold exposure-induces hypothalamic miR-33 via ER stress is wick as no experiments performed by the authors directly support this conclusion. They showed that while 6 hours cold exposure induced Bip and Chop gene expressions, it takes 7 days of cold exposure to induce miR33. Furthermore, while thapsigargin treatment induces Bip and Chop expression only at a concentration of 0.1uM, it takes a much lower dose of 0.01uM to induce miR-33.

Thank you very much for these critical and valuable comments. We performed western blotting for BIP, CHOP, and cleaved-caspase 3 in Fig. 3 (Fig. 4 in the revised manuscript). In the experiment with Neuro2a cells, a high concentration of thapsigargin (0.1 μ M) induced the expression of BIP, CHOP, and cleaved-caspase 3. On the other hand, a low concentration of thapsigargin (0.01 μ M) also induced BIP and CHOP without an increase in cleaved-caspase 3 (Fig. 4f and 4g). Because miR-33 elevation was more prominent with a low thapsigargin concentration (0.01 μ M) than a high concentration (0.1 μ M), mild ER stress without cell death may be enough and efficient for miR-33 induction. In the hypothalamus, ER stress markers *Bip* and *Chop* were mildly induced by cold exposure for 6 h, and these inductions were reversed by subsequent

exposure to room temperature for 18 h (Fig. 4b). Therefore, ER stress evoked by cold stress is also considered to be mild and weaker than that of thapsigargin. Thus, the difference in the duration for miR-33 elevation between *in vitro* and *in vivo* studies may be due to the degree of ER stress.

We added and revised the description in the main text as follows.

•Lines 241–252 in the revised manuscript;

“The expression levels of ER stress markers *Bip* and *Chop* in the hypothalamus were mildly induced by cold exposure for 6 h, and these inductions were reverted by subsequent exposure to room temperature for 18 h (Fig. 4b). This BIP induction was also confirmed at the protein level by western blotting (Fig. 4c and 4d). Cleaved-caspase 3, one of the markers of apoptosis, was not detectable in this condition. In an *in vitro* study, we treated Neuro2a cells with thapsigargin, an ER stress inducer, and found that *Srebf2* and miR-33 were induced along with *Bip* and *Chop* even at a low concentration (Fig. 4e). Significant inductions of BIP and CHOP protein levels were observed at a high concentration of thapsigargin (0.1 μ M), which were accompanied with an increased level of cleaved-caspase 3 (Fig. 4f and 4g). On the other hand, a low concentration of thapsigargin (0.01 μ M) also significantly induced BIP and CHOP but without an increase in cleaved-caspase 3 (Fig. 4f and 4g). Because the miR-33 increase was prominent at a low concentration, mild ER stress may be sufficient for miR-33 induction.”

•Lines 443–448 in the revised manuscript;

“The experiments with Neuro2a cells suggest that mild ER stress without apoptosis may be sufficient for miR-33 elevation. In the hypothalamus, ER stress markers were also mildly elevated in a cold stress condition. In fact, although a significant BIP induction was observed, an increase in cleaved-caspase 3 was not observed in this condition, which indicates that ER stress evoked by cold stress is mild. The difference in the duration for miR-33 elevation between *in vitro* and *in vivo* may be due to the degree of ER stress.”

2. In addition, the critical measurement of intrascapular BAT temperature is missing in the entire study in all animal models used. This is an important measurement especially when not always the markers of thermogenesis in the BAT are significantly different or when the difference between the experimental groups is marginal.

Thank you very much for pointing out this important issue. Based on the suggestion, we directly measured BAT temperature using a needle-type probe and thermometer along with measurement of rectal temperature in miR-33^{+/+} and miR-33^{-/-} mice. When anesthetized mice were placed in a cold environment, both BAT and rectal temperature declined. However, BAT temperature declined more slowly than rectal temperature, which indicate that BAT thermogenesis occurs. Decline in BAT temperature was more prominent and the difference in BAT and rectal temperature was smaller in miR-33^{-/-} mice than miR-33^{+/+} mice. These data indicate that BAT thermogenesis is impaired in miR-33^{-/-} mice. These data are shown in Fig. 2n.

We added and revised the description in the main text as follows.

•Lines 178–182 in the revised manuscript;

“Moreover, we measured both BAT and rectal temperature at the same time in a cold condition. BAT temperature declined more slowly than rectal temperature, which indicates that BAT thermogenesis was occurring. Decline in BAT temperature was more prominent and the difference in BAT and rectal temperature was smaller in miR-33^{-/-} mice than miR-33^{+/+} mice (Fig. 2n).”

•Lines 573–576 in the revised manuscript;

“Measurement of BAT and rectal temperature; anesthetized mice were placed on ice, and BAT and rectal temperature were recorded every minute at the same time. A needle-type probe and thermometer was used for the direct measurement of BAT temperature (AD-1220-100 and AD5601A) (A & D company, Japan).”

3. In all figures showing Western blots, graphs showing results from the densitometry analyses should be reported. This is true for Fig. 1e, 2f, 4e, 5g.

In Fig1i, the labeling of the X axis is wrong. miR33^{-/-}(23°C dark) should be miR33^{-/-}(18°C dark).

Thank you very much for this important comment and pointing out this error. Based on the comments, we performed densitometric analysis for UCP1 in Fig. 1e (Fig. 1f in the revised manuscript), 2f (Fig. 3f in the revised manuscript), 4e (Fig. 5e in the revised manuscript), and 5g (Fig. 6g in the revised manuscript)

and have shown the results in Supplementary Fig. 3d, Supplementary Fig. 5d, Supplementary Fig. 6b, and Supplementary Fig. 7l, all of which show statistically significant differences. We also corrected the label in Fig. 1i (Fig. 2i in the revised manuscript).

4. In addition, the experiment in Fig1j is confusing. First NE administration (which is not described in the methods section) does not seem to increase oxygen consumption in the control if one compares the graph in Fig 1i with the graph in Fig 1j. Then, I would expect that NE administration would have a strong effect in the KO mice in which the activation of the SNS is lower. It would have been more informative to administer a dose of NE that was not effective in the controls hoping to show that the same dose had an effect in the KO.

Thank you very much for these comments. Although mean oxygen consumption rate was measured at 18°C in Fig. 1i (Fig. 2i in the revised manuscript), maximum oxygen consumption was measured at thermoneutrality in Fig. 1j (Fig. 2m in the revised manuscript). In addition, these mice were anesthetized; therefore, the metabolic rate at baseline was low and the augmentation by norepinephrine can be seen depending on the expression levels of UCP1 at that time. Details of this experiment are described as follows in this literature (Cannon, B. & Nedergaard, J. *J Exp Biol* 214, 242–53 (2011)); “Because it is norepinephrine that activates nonshivering thermogenesis, one means to evaluate its capacity of an animal is to treat it acutely with norepinephrine to mimic activation of the sympathetic nervous system. The magnitude of the response will vary, depending on the previous condition of mice. It is important that the norepinephrine injection experiments are performed at thermoneutrality to avoid BAT activation and ‘paradoxical’ effects of norepinephrine, which are often seen at low temperature. To improve the reproducibility of the measurements and decrease the number of animals required, anaesthetized animals can be studied.”

We previously described how to measure maximum oxygen consumption in the sub-section entitled “Assessment of metabolic rate” in the Methods section of our original manuscript as follows;

“For the assessment of maximum oxygen consumption rate, male mice kept at the indicated temperature for 3 days were anesthetized with pentobarbital, and then transferred to a chamber at 30°C. After measurement of basal rates for 30 min, mice were subcutaneously injected with norepinephrine (1 mg/kg), and oxygen consumption rate was measured for another 60 min.”

In the revised version, we separated this method from the previous sub-section and described in a new sub-section entitled “Assessment of maximum oxygen consumption rate”. (Lines 584–588 in the revised manuscript).

5. According to Figure 4a and b, miR-33b^{+/+} mice showed significant increase in body temperature compared to miR-33b^{-/-} mice. However, in supplementary Figure 6g, body temperature in these same animal groups at time 0 is not significant. How do the authors reconcile this difference?

Thank you very much for pointing out this difference in our data. We considered that the difference were due to the smaller number of mice analyzed. We added data from 3–4 additional mice and re-analyzed the data. We revised Supplementary Fig. 6l as a new graph of n=12, 13 mice. As a result, at 0 h, 1 h and 3 h point, the body temperature was significantly higher in miR-33b^{+/+} mice. We also revised the q-PCR data of this experiment in Supplementary Fig. 6m.

We revised the description in the main text as follows.

•Lines 277–280 in the revised manuscript;

“In a cold condition, miR-33b^{+/+} mice ameliorated body temperature decline and showed increased expression of thermogenic genes including *Ucp1*, *Ppargc1a*, and *Pparg* in the BAT. (Supplementary Fig. 6l and 6m).”

6. In Figure 6b, miR-33a/b double knockout (KO) human iPS cells showed significant up-regulation of GABRB2 and GABRA4. However, single KOs showed no significant changes in GABRB2 and GABRA4 expressions suggesting that both miR-33a/b are required for gene regulation of GABRB2 and GABRA4. Since the mouse model generated overexpresses only miR-33b, how do the authors explain their results and the effect of the shRNA targeting *Gabrb2* and *Gabra4* (Figure 6g-j)? Could this be miR-33-independent? This should be considered especially because of the

**approach used to induce the downregulation of Gabrb2 and Gabra4
shRNA AAV9 infection of miR-33f/f DBH-cre mice.**

Thank you very much for raising this important question. Rodents have only one miR-33 in an intron in the *Srebf2* gene, whereas humans have two miR-33s, miR-33a in an intron of the *SREBF2* gene and miR-33b in the intron of *SREBF1* gene. Because both miR-33a and miR-33b target GABRB2 and GABRA4, deletion of both miR-33a and miR-33b is necessary to change in the expression of these genes in human cells including human iPS cells. In contrast, mice have only one miR-33 in general (corresponding to miR-33a in human cells), deletion of miR-33 is sufficient to see changes in these target genes. It is also true in miR-33^{f/f} *DBH-Cre* mice, which have only one miR-33 in *DBH*-negative cells and no miR-33 in *DBH*-positive cells. In these mice, the target genes are expected to be increased in *DBH*-positive cells. Because shRNA is expressed only in cells that express Cre recombinase, the increased expression of target genes in *DBH-Cre* positive cells are reverted by conditional shRNA AAV9 infection. Thus, the phenotypes observed by the rescue experiments are considered as miR-33-dependent ones. Regarding miR-33b, it is impossible to define the function of miR-33b in mice, except by using the humanized miR-33b knock-in (miR-33b^{+/+}) mice that have both miR-33a and miR-33b.

Response to Reviewer #3

We are grateful to reviewer #3 for the important and critical comments and useful suggestions that have helped us to improve our manuscript considerably. As indicated in the responses that follow, we have taken all these comments and suggestions into account in the revised version of our manuscript.

This study by Horie et al seeks to understand how miR-33, a well-known miRNA that has been reported to control metabolism and obesity, contributes to whole-body metabolism via its role in brown adipose tissue. Many previous studies have shown that loss of miR33 causes weight gain in the setting of high fat diet, and the authors seek in this study to better determine the cause of these observations. They use loss- and gain-of-function approaches in male mice, and use a Cre-mediated deletion of miR-33 using dopamine- β -hydroxylase Cre to delete miR33 in catecholamine-producing cells. They examine sympathetic innervation as a major mechanism controlling the observed defects in thermogenesis in miR-33^{-/-} mice, and show that miR33 controls the expression of GABAergic genes *Gabra2* and *Gabra4* in the hypothalamus.

This study is interesting and suggests novel mechanisms by which miR-33, which has previously been characterized mainly as a therapeutic target for cardiometabolic diseases, controls thermogenesis via activity in specific brain regions that control brown adipose tissue energy utilization.

However, the conclusions in this study are not supported by the data, largely because the measurements were taken from different time points, across different exposures to cold temperatures, and many of the differences between groups are not quantified and appear small in effect size. While the use of the DBH-Cre mice are interesting, they confound the study because of their lack of specificity and the lack of characterization of the other cell(s) that may be influenced by miR33 deletion.

Major comments:

1. The authors conclude that in the 2-week HFD study, there is “attenuation” of Ucp-1 expression with miR33^{-/-}. This is not well supported by the data shown in the paper. First, the UCP-1 IHC in Supp Fig 2 needs to be more clear and both the staining and the WB need to be quantified. In the image shown in the figure, there does not appear to be an appreciable reduction

in Ucp1 protein. This is also true for many of the Western blots in the manuscript, which do not show quantification.

Thank you very much for these valuable comments. We agree with the point by the reviewer. We replaced the images of IHC for UCP1 in Supplementary Fig. 2f with magnified ones. We also quantified the UCP1-positive area in IHC, which is now shown in Supplementary Fig. 2g. Although the UCP1-positive areas were increased by HFD feeding in miR-33^{+/+} control mice, the increase was attenuated in miR-33^{-/-} mice.

We also quantified UCP1 levels in Supplementary Fig. 2c, and the results are shown in Supplementary Fig. 2d (n=4).

We further performed densitometric analysis of UCP1 in Fig. 1f, 3f, 5e, and 6g and showed the results in Supplementary Fig. 3d, Supplementary Fig. 5d, Supplementary Fig. 6b, and Supplementary Fig. 7l, all of which showed statistically significant differences.

2. There is a discordance in the data regarding timing of the analyses. For example in Figure 1, some analyses are done after 6h of cold exposure, others after 48h. Please show protein expression after 6h, as this is purported to be when miR-33 is exerting physiologic effects (according to Fig 1a-c). Similarly, OCR was measured after 3 days at 18°C, yet gene expression, protein expression, BAT lipid droplets etc were not.

Thank you very much for these critical and valuable comments. We performed western blotting of UCP1 after cold exposure for 6 h in BAT; however, UCP1 induction at the protein levels was not observed at this time point (data shown below). From the current study, we found that activation of the sympathetic nerve system (SNS) itself is different between miR-33^{+/+} and miR-33^{-/-} mice. Therefore, the phenotype of the BAT at 6 h is considered to have originated from the difference in the activation of the SNS responding to cold exposure. Because UCP1 expression in the BAT is under the control of the SNS, we further checked the expression of UCP1 sequentially and confirmed that significant increases in UCP1 were observed after 30 h and 48 h of cold exposure (data shown below and Fig. 1f), thus we showed the OCR data in a relatively later phase. With regard to the OCR in cold exposure, we presented the data after 3 days of cold

exposure in the original version, because the variation in the values at 3 days was smaller than after 2 days. However, statistical analysis showed that the change was also significant in the data for OCR after both 2 days and 3 days of cold exposure. We revised the OCR data including VO_2 and other parameters after 2 days at 18°C in Fig. 2g-2l, which were almost the same as after 3 days at 18°C. We also showed a picture of the BAT after 2 days of cold exposure in Fig. 1e and Supplementary Fig. 3b. The amount of lipid droplets remained significantly higher in miR-33^{-/-} mice in this condition (Supplementary Fig. 3c).

We revised the description in the main text as follows.

•Lines 169-173 in the revised manuscript;

“After 2 days of cold exposure at 18°C, the difference in the oxygen consumption became evident and significantly reduced in miR-33^{-/-} mice in the dark phase (Fig. 2g and 2i). The carbon dioxide production was also significantly reduced in miR-33^{-/-} mice (Fig. 2h and 2j). The RER and locomotor activity were not different between miR-33^{+/+} and miR-33^{-/-} mice in this condition (Fig. 2k and 2l).”

3. There are no measures of BAT fuel oxidation or electron transport chain activity, which is a primary mechanism that promotes thermogenesis and are downstream of Ucp1. The authors need to demonstrate that miR-33 impacts the uncoupling process in BAT.

Thank you very much for your important comment. We measured FA oxidation (FAO) activity in the BAT by a non-radiolabeled enzymatic assay kit, using homogenates of BAT. Although FAO activity was similar at room temperature, it was significantly reduced in miR-33^{-/-} mice at 2 days after cold exposure. These data is shown in Supplementary Fig. 3g

We added and revised the description in the manuscript as follows.

•Line 173-175 in the revised form;

“Fatty acid oxidation activity in the BAT was significantly reduced in miR-33^{-/-} mice in a cold environment (Supplementary Fig. 3g).”

•Line 590-596 in the revised form;

“Measurement of fatty acid oxidation activity; we measured fatty acid oxidation (FAO) activity in the BAT using a non-radioactive enzymatic assay kit according to the instruction (BMR, NY)⁶⁰. The measurement is based on the oxidation of octanoyl-CoA, which is coupled to NADH-dependent reduction of INT to INT-formazan. The formazan production from the same amount of BAT homogenates with or without octanoyl-CoA were measured by ARVO X3 microplate reader at O.D. 492 nm (PerkinElmer, MA). The subtracted O.D. values are proportional to the FAO activity and presented.”

4. The denervation experiments presented in Figure 2 are interesting and support that miR33 requires sympathetic nerve activity to stimulate thermogenic genes. However, given that BAT sympathetic activity is defective after denervation and unable to stimulate thermogenesis, it is important to show negative and positive control genes that are not controlled by denervation, as well as ones that are not under the control of miR-33 in order to conclude that these changes are due to loss of miR33.

Thank you very much for pointing out the importance of positive and negative controls for this experiment. *Ucp1* and *Pgc1a* (*ppargc1a*) are positive controls for denervation, because they are under the control of the sympathetic nerve system. As negative controls for denervation, we checked several house-keeping genes including β 2 microglobulin (*B2m*), *36b4*, and *Gapdh*. The expression levels of *B2m* and *36b4* were similar between sham and denervation, or miR-33^{+/+} and miR-33^{-/-} mice, which can be considered as negative controls for denervation. On the other hand, *Gapdh* levels were decreased by denervation, and its expression was similar between miR-33^{+/+} and miR-33^{-/-} mice, which can be considered as a positive control that is not under the control of miR-33. These results are shown in Supplementary Fig. 5c.

We added and revised the description in the main text as follows.

•Lines 212–213 in the revised manuscript;

“The expression levels of several internal control genes including *B2m*, *36b4*, and *Gapdh* are shown in Supplementary Fig. 5c.”

5. Why do TH levels change in Figure 2f (WAT) but not in Figure 1e (BAT)? The authors do not discuss this yet the loss appears dramatic. These data confound whether miR-33 loss impairs denervation of the BAT or not.

Thank you very much for your valuable suggestion. When we performed western blotting using an antibody against TH (AB152, Millipore), two bands could be seen. The lower, stronger bands are TH-specific bands, which are indicated by arrow in the figures. Densitometry of TH in BAT (Fig. 1f) did not show any difference, which is shown in Supplementary Fig. 3d. In the original version of TH staining in Inguinal WAT (Fig. 2f), the rightmost side lane seemed to be extremely reduced, whereas the others were not. It seemed that there was an uneven staining of the blotting. We performed the western blotting analysis again and replaced with the new data in Fig. 2f (Fig. 3f in the revised manuscript). Densitometric analysis showed a significant reduction in UCP1 levels; however, TH levels were similar between miR-33^{+/+} and miR-33^{-/-} mice, which is shown in Supplementary Fig. 5d. We also quantified all of the other TH immunoblotting. Densitometric analysis of TH in BAT of miR-33b^{+/+} mice (Fig. 5e), Inguinal WAT in miR-33^{fl/fl} DBH-Cre mice (Fig. 6g), and BAT in miR-33^{-/-} mice with HFD feeding (Supplementary Fig. 2c) are shown in Supplementary Fig. 6b, 7l, and 2d, all of which did not show significant differences.

6. The gene expression analyses in the hypothalamus in Figure 3 are done at different time points and after different patterns of cold exposure of the mice. Therefore, conclusions about miR33 levels and ER stress markers cannot be drawn, nor can mechanisms be extrapolated from the previous data in miR33^{-/-} mice as the cold exposure protocol was different and similar time points are not examined.

Thank you very much for these critical and important comments. In the experiment with Neuro2a cells, a high concentration of thapsigargin (0.1 μM) induced the expression of BIP, CHOP, and cleaved-caspase 3 (Fig. 4f and 4g). On the other hand, a low concentration of thapsigargin (0.01 μM) also induced

BIP and CHOP without an increase in cleaved-caspase 3 (Fig. 4f and 4g). Because miR-33 elevation was more prominent with a low concentration of thapsigargin (0.01 μ M) than a high concentration (0.1 μ M) (Fig. 4e), mild ER stress without cell death may be enough and efficient for miR-33 induction. In the hypothalamus, ER stress markers *Bip* and *Chop* were mildly induced by cold exposure for 6 h, and these inductions were reversed by subsequent exposure to room temperature for 18 h (Fig. 4b). Therefore, ER stress evoked by cold stress was also considered to be mild. Because miR-33 expression was not significantly changed by cold exposure for 6 h, we thought that ER stress evoked by cold stress was weaker than that of thapsigargin, and the difference in the duration for miR-33 elevation between *in vitro* and *in vivo* analyses might be due to the degree of ER stress. Therefore, we tried to repeat the cold stimulation with mice. When cold exposure for 6 h/day was repeated, some mice did not survive. Thus, we reduced the duration of cold exposure from 6 h to 3 h and increased the number of days of treatment to enhance the effect of ER stress. As a result, the induction of hypothalamic miR-33 occurred in a relatively later phase, which likely contributed to chronic adaptation to the cold environment.

We added and revised the description in the main text as follows.

•Lines 241–252 in the revised manuscript;

“The expression levels of ER stress markers *Bip* and *Chop* in the hypothalamus were mildly induced by cold exposure for 6 h, and these inductions were reverted by subsequent exposure to room temperature for 18 h (Fig. 4b). This BIP induction was also confirmed at the protein level by western blotting (Fig. 4c and 4d). Cleaved-caspase 3, one of the markers of apoptosis, was not detectable in this condition. In an *in vitro* study, we treated Neuro2a cells with thapsigargin, an ER stress inducer, and found that *Srebf2* and miR-33 were induced along with *Bip* and *Chop* even at a low concentration (Fig. 4e). Significant inductions of BIP and CHOP protein levels were observed at a high concentration of thapsigargin (0.1 μ M), which were accompanied with an increased level of cleaved-caspase 3 (Fig. 4f and 4g). On the other hand, a low concentration of thapsigargin (0.01 μ M) also significantly induced BIP and CHOP but without an increase in cleaved-caspase 3 (Fig. 4f and 4g). Because the miR-33 increase was prominent at a low concentration, mild ER stress may be sufficient for miR-33 induction.”

•Lines 443–450 in the revised manuscript;

“The experiments with Neuro2a cells suggest that mild ER stress without apoptosis may be sufficient for miR-33 elevation. In the hypothalamus, ER stress markers were also mildly elevated in a cold stress condition. In fact, although a significant BIP induction was observed, an increase in cleaved-caspase 3 was not observed in this condition, which indicates that ER stress evoked by cold stress is mild. The difference in the duration for miR-33 elevation between in vitro and in vivo may be due to the degree of ER stress. Taken together, cold stress or HFD feeding induces mild ER stress in the hypothalamus, in turn, ER stress can up-regulate the expression of Srebf2 and its intronic miR-33.”

7. The data from miR33b^{+/+} mice are not consistent. First, there is a large difference in the number of animals analyzed in Figures 4 and Supplemental Fig 6. Second, there are few if any changes in gene expression of thermogenic genes at 4 degrees in miR33b mice compared to controls (only room temperature shows differences in expression).

Thank you very much pointing out the difference in our data. We measured the body temperature of miR-33b^{-/-} and miR-33b^{+/+} mice on different days using different sets of mice at room temperature (three times in males, two times in females). In each measurement, a significant elevation of the body temperature in miR-33b^{+/+} mice was observed, as shown below. We combined all of the data and presented it in Fig. 4a and b (Fig. 5a and 5b in the revised manuscript), as a result, the total number of mice was increased.

In males:

- 1st measurement: 35.57±0.16 vs 36.94±0.13 (n=7, 7 each p=0.0021, Mann–Whitney test)
- 2nd measurement: 36.81±0.17 vs 37.58±0.16 (n=12, 9 each p=0.0060, Mann–Whitney test)
- 3rd measurement: 36.46±0.20 vs 37.17±0.11 (n=5, 9 each p=0.0032, Mann–Whitney test)
- Total: 36.38±0.15 vs 37.25±0.10 (n=24, 25 each p<0.0001, Mann–Whitney test)

In females:

- 1st measurement: 35.47±0.12 vs 36.04±0.15 (n=11, 9 each p=0.0067, Mann–Whitney test)

- 2nd measurement: 35.88±0.08 vs 36.75±0.06 (n=21, 22 each p<0.0001, Mann–Whitney test)
- Total: 35.74±0.07 vs 36.54±0.08 (n=32, 31 each p<0.0001, Mann–Whitney test)

We considered that the difference were due to the smaller number of mice analyzed in the experiment with cold exposure of miR-33b^{+/+} mice. We added the data of 3–4 more additional mice and re-analyzed the data. We revised Supplementary Fig. 6l as a new graph with n=12, 13 mice. As a result, at the 0 h, 1 h, and 3 h time points, the body temperature was significantly higher in miR-33b^{+/+} mice. We also re-analyzed gene expression in the BAT and revised the q-PCR data from this experiment in Supplementary Fig. 6m. The difference in *Ucp1* and *Pparg* reached statistical significance, along with *Pparg1a*.

We revised the description in the main text as follows.

Lines 277–280 in the revised manuscript;

“In a cold condition, miR-33b^{+/+} mice ameliorated body temperature decline and showed increased expression of thermogenic genes including *Ucp1*, *Pparg1a*, and *Pparg* in the BAT. (Supplementary Fig. 6l and 6m).”

8. According to the MGI, the DBH-Cre mouse model used in this system “Cre activity is detected in motor neurons of the cranial nerves and spinal cord, and sensory and enteric neurons”. The authors therefore need to verify locomotor activity and other motor neuron features to ensure that miR-33 loss is not impacting whole-body motor function.

Thank you very much for these valuable suggestions and advice. We collected and analyzed the locomotor activity from the recorded data while in the metabolic cages. In this analysis, the number of times that infrared beams were broken by mice during the experiment were recorded and counted. We added locomotor activity data of miR-33^{-/-} mice at room temperature (Fig. 2f) and after cold exposure (Fig. 2l) and the locomotor activity data of miR-33b^{+/+} mice at room temperature (Supplementary Fig. 6g). Moreover, we added locomotor activity data from miR-33^{fl/fl} DBH-Cre mice after cold exposure (Supplementary Fig. 7k). The results were similar compared with littermate controls. Because

locomotor activity was not affected by miR-33 modulation, we concluded that loss of miR-33 had less impact for whole-body motor function.

We added the following description in the main text.

•Lines 168–173 in the revised manuscript;

“The respiratory exchange rate (RER) and locomotor activity were similar between miR-33^{+/+} and miR-33^{-/-} mice (Fig. 2e and 2f). After 2 days of cold exposure at 18°C, the difference in the oxygen consumption became evident and significantly reduced in miR-33^{-/-} mice in the dark phase (Fig. 2g and 2i). The carbon dioxide production was also significantly reduced in miR-33^{-/-} mice (Fig. 2h and 2j). The RER and locomotor activity were not different between miR-33^{+/+} and miR-33^{-/-} mice in this condition (Fig. 2k and 2l).”

•Lines 270–271 in the revised manuscript;

“The RER and locomotor activity were similar between miR-33b^{-/-} and miR-33b^{+/+} mice (Supplementary Fig. 6f and 6g).”

•Lines 307–308 in the revised manuscript;

“The carbon dioxide production, RER, and locomotor activity were similar between miR-33^{ff} and miR-33^{ff} *DBH*-Cre mice (Supplementary Fig. 7i-7k).”

9. Similar to the above comment, according to the data in Supp Figure 4, skeletal muscle expresses a relatively high level of miR-33. Skeletal muscle also participates in thermogenesis and thus may be contributing to the role of miR33 in this regulation. In the studies of whole body metabolism of miR33^{-/-} mice, the authors should show physical activity measurements and characterize the skeletal muscle in response to cold exposure.

Thank you very much for these valuable comments. As described above, we analyzed locomotor activity from the data recorded from the metabolic cages. As a result, the locomotor activity level was not different between miR-33^{+/+} and miR-33^{-/-} mice (Fig. 2f). When we compared locomotor activity at room temperature (Fig. 2f) with that after cold exposure (Fig. 2l), the results were also similar between the comparisons. Therefore, cold exposure does not alter their physical activity or skeletal muscle function regarding locomotor activity, which may not contribute to whole-body metabolism.

10. It is concerning that miR-33 levels were not significantly different in DBH-Cre mice according to Supp Figure 7e (except for in the adrenal gland). The authors justify the choice of the DBH-Cre system as this deletes floxed genes in “catecholamine-producing cells like noradrenergic and adrenergic cells [...] including several hypothalamic nuclei such as the DMH”. There is a trend but no significant changes in miR33 expression in the hypothalamus. This region is also responsible for food intake, which is not accounted for anywhere in this study, yet should be examined in both DBH-Cre and whole-body miR33^{-/-} mice.

Thank you very much for these important comments. We thought that it was due to the smaller number of mice analyzed. We increased the number of mice for analysis and re-analyzed the data (Mann–Whitney test). The expression of miR-33 in the hypothalamus ($p=0.0079$) and adrenal gland ($p=0.0317$) were significantly reduced in miR-33^{ff} DBH-Cre mice. We also determined the statistical differences for other organs. However, they had not exhibit significant differences (cortex ($p=0.0556$), BAT ($p=0.2222$), WAT ($p=0.6905$), and liver ($p=0.4206$)). We added a new graph in Supplementary Fig. 7e. Additionally, we have shown data on food intake by miR-33^{-/-} and miR-33^{ff} DBH-Cre mice in Supplementary Fig. 1b and Supplementary Fig. 7m, respectively. Although food intake was significantly higher in miR-33^{-/-} mice than miR-33^{+/+} mice, as reported previously, it was similar between miR-33^{ff} and miR-33^{ff} DBH-Cre mice unlike miR-33^{-/-} mice. These data indicate that DBH-positive neurons are not responsible for increased food intake in miR-33^{-/-} mice.

We added and revised the description in the main text as follows.

•Lines 295–297 in the revised manuscript:

“The expression levels of miR-33 in the hypothalamus and adrenal glands were significantly reduced in miR-33^{ff} DBH-Cre mice (Supplementary Fig. 7e).”

•Lines 119-120 in the revised manuscript:

“Food intake of an HFD was slightly but significantly higher in miR-33^{-/-} mice, as reported previously (Supplementary Fig. 1b).”

•Lines 314–318 in the revised manuscript:

“To further characterize the metabolic phenotype of miR-33^{ff} DBH-Cre mice, we fed an HFD to these mice. Unlike miR-33^{-/-} mice, the amounts of food intake

were similar between miR-33^{fl/fl} mice and miR-33^{fl/fl} *DBH-Cre* mice, which indicates that loss of miR-33 in *DBH*-positive neurons is not responsible for the increased food intake in miR-33^{-/-} mice (Supplementary Fig. 7m and 1b).”

•Lines 502–505 in the revised manuscript:

“On the other hand, food intake is another important factor for obesity. It was increased in miR-33^{-/-} mice but not in miR-33^{fl/fl} *DBH-Cre* mice, which indicates that miR-33 in *DBH*-positive neurons is not responsible for the increased food intake in miR-33^{-/-} mice. Further analysis is needed to elucidate this mechanism.”

Minor points:

1. In the short-term diet feeding study, why did they switch from 45% to using 60% HFD?

Thank you very much for your query on our data. To increase the burden of HFD feeding in as little as 2 weeks, we chose 60% HFD in the short-term diet feeding study.

2. The axis on Figure 1a is compressed, and there are no error bars present. These should be shown.

Thank you very much for this suggestion. Because there was less variation in the data, error bars were difficult to be distinguished. When we de-compressed the axis, error bars became even smaller. We made the plots smaller in the Fig. 1a in the revised manuscript, although the error bars at 0–3 h cannot be seen because of their lower variation.

Response to Reviewer #4

We are grateful to reviewer #4 for the positive comments and valuable suggestions that have helped us to improve our paper considerably. As indicated in the responses that follow, we have taken all these comments and suggestions into account in the revised version of our paper.

This is an excellent manuscript detailing the role of diminished BAT thermogenesis in the reduced cold defense and increased susceptibility to HFD in miRNA-33^{-/-} mice. The authors have significantly extended their study to demonstrate that these metabolic alterations arise from an increased GABAA-R expression in catecholamine neurons in the brain, resulting in a reduced sympathetic outflow to BAT. These results point to an important role for the miRNA-33 regulation of the GABAA-R expression in BAT regulating networks in the CNS. There is wide scientific interest in, and increasing appreciation of the roles of miRNAs in controlling the activity of a variety of tissues, and the high quality of this research will ensure that this report is highly cited.

1. There is no quantification for either the BAT staining data concerning lipid droplets or the gel data on protein levels. Since the authors describe significant differences in these measures, quantification and statistics should be provided.

Thank you very much for these valuable suggestions. We performed quantification of the lipid droplet area in the BAT of miR-33^{-/-} mice at room temperature or after cold exposure for 6 h and 48 h, miR-33b^{+/+} mice at room temperature, miR-33^{fl/fl} *DBH-Cre* mice after cold exposure, and miR-33^{fl/fl} *DBH-Cre* mice infected with shRNA AAV9. All of the data showed that there were significant differences compared with littermate controls or shRNA control-infected control mice. These data are shown in Supplementary Fig. 3a, Supplementary Fig. 3c, Supplementary Fig. 6a, Supplementary Fig. 7f, and Supplementary Fig. 9i. We also performed densitometric analysis for UCP1 in Fig. 1f, Fig. 3f, Fig. 5e, Fig. 6g, and Supplementary Fig. 2c and showed the results in Supplementary Fig. 3d, Supplementary Fig. 5d, Supplementary Fig. 6b, Supplementary Fig. 7l, and Supplementary Fig. 2d, all of which showed

statistically significant differences. TH levels on the same blots were similar between the comparisons.

2. Lines 128, 148, 164, 168 – the authors describe “trends” in their data, but this is not statistically valid: if there was no significant difference, then the data should be reported as “not different”, rather than the biased assessment of “trending” in the direction the authors would like.

Thank you very much for these valuable comments. We agree with these suggestions and revised lines 128, 148, 164, and 168 as follows.

·Line 128: “Gene expression levels of thermogenic genes including *Ucp1* in the BAT were not significantly reduced in these mice (Supplementary Fig. 1h)” (Lines 127–129 in the revised manuscript)

·Line 148: “Lipid droplets in the BAT were slightly but significantly larger in miR-33^{-/-} mice than miR-33^{+/+} mice at room temperature, which is a mild cold situation for mice (Fig. 1b and Supplementary Fig. 3a).” because it was statistically significant by quantification of lipid droplets as mentioned above (Supplementary Fig. 3a). (Lines 148–150 in the revised manuscript)

·Line 164: “The oxygen consumption rate at 23°C (room temperature) was not significantly different between miR-33^{+/+} and miR-33^{-/-} mice (Fig. 2a and 2c).” (Lines 165–166 in the revised manuscript)

·Line 168: We deleted the following sentence; “, which had a slight tendency towards a reduction in miR-33^{-/-} mice.” (Lines 175–176 in the revised manuscript).

3. As the authors are likely aware, “room temperature” (line 153) is actually a mild cold exposure for the mouse and should be described in this way. Exposure to 4C should be described as “deep cold”, keeping in mind that the threshold for cold nociceptors is around 15C.

Thank you very much for these valuable comments. Based on this comment, we added explanation of the findings at room temperature, with ‘mild cold exposure’, and at 4°C with ‘deep cold exposure’.

We revised the description in the main text as follows.

·Lines 145–147, 148–152 in the revised manuscript;

“To evaluate cold-induced adaptive thermogenesis, we placed 8-weeks-old mice in a 4°C deep cold environment and measured serial core body temperature.”
“Lipid droplets in the BAT were slightly larger in miR-33^{-/-} mice than miR-33^{+/+} mice at room temperature, which is a mild cold situation for mice (Fig. 1b and Supplementary Fig. 3a). After a deep cold exposure (4°C), lipid droplets were dramatically reduced in miR-33^{+/+} mice, whereas they were still present in miR-33^{-/-} mice (Fig. 1c and Supplementary Fig. 3a).”

4. Although they probably didn't make any relevant observations, the authors might consider a brief mention in their Discussion of the potential for a similar regulation by miR-33 of shivering thermogenesis. The CNS pathways for shivering and for BAT thermogenesis are nearly overlapping (premotor neurons in rRPa excited by their antecedent neurons in the DMH), thus it is likely that miR-33 is influencing shivering in a manner similar to its regulation of BAT thermogenesis. Along these lines, shivering is a much more significant source of energy expenditure in humans than is BAT, thus it may be more relevant for weight loss in humans.

Thank you very much for these valuable comments. Although we did not examine shivering thermogenesis in this manuscript, we fully agree with the reviewer's opinion.

We added and revised the description in the main text as follows.

•Lines 430–434 in the revised manuscript;

“Moreover, shivering thermogenesis, which was not examined in this study, is also important for thermogenesis in cold environments. Because regulation of the central nervous system for shivering thermogenesis is known to be similar to that of BAT thermogenesis⁶, it might be also changed by miR-33 alternation in the brain and affect the maintenance of body temperature and whole-body metabolism.”

5. Lines 101-102 must be reworded to “In the current study, we analyzed several miR-33 genetically modified mice and found that miR-33 maintains BAT thermogenesis.”

Thank you very much for this kind suggestion. We reworded the sentence as suggested

·Lines 100–101 in the revised manuscript;

“In the current study, we analyzed several miR-33 genetically modified mice and found that miR-33 maintains BAT thermogenesis.”

6. Lines 204-205 (and elsewhere): NETO differences could arise for a variety of reasons, including differences in the ongoing level of sympathetic outflow, but also if there are different numbers of ganglion cells or different numbers of sympathetic axon terminals, the latter being an active research area as new growth factor molecules are being identified that are secreted by BAT and alter the density of its sympathetic innervation. Thus, the NETO data should be described as a potential surrogate for sympathetic nerve activity, but it is not a direct substitute for it.

Thank you very much for these valuable comments. We added an explanation to the NETO experiments in the Results section as follows.

·Lines 215–218 in the revised manuscript;

“Next, we measured norepinephrine turnover (NETO) in BAT using α -methyl-DL tyrosine (AMPT), which is an inhibitor of tyrosine hydroxylase that is a rate-limiting enzyme for NE synthesis, because it is used as a potential surrogate for sympathetic nerve activity^{25,28}.”

7. Lines 233-234 (and elsewhere): although it is likely that ER stress induces miR-33, the authors have not demonstrated that the increased miR-33 with HFD or cold results from ER stress, but rather simply that both occur together. They would need to show that preventing ER stress during HFD or cold also prevents the induction of miR-33. Thus, this sentence (and similar conclusions elsewhere, including line 464) should be reworded to eliminate “indicate”, perhaps replacing with “suggest”.

Thank you very much for these useful comments. We reworded “indicate” to “suggest” in Lines 233–234 (Lines 254–255 in the revised manuscript);

“These data suggest that cold exposure or HFD feeding induces miR-33 expression via increased ER stress in the hypothalamus.”

We also add “possibly” in front of “via ER stress” in Lines 365–366 (Lines 401–402 in the revised manuscript) and Lines 463–466 (Lines 507–510 in the revised manuscript);

“Furthermore, we found that cold stress up-regulated miR-33 levels in the hypothalamus possibly via ER stress.”

“During cold stress miR-33 in the brain, in particular the hypothalamus, which is up-regulated possibly via ER stress, contributes to the maintenance of BAT thermogenesis and whole-body metabolism through increased sympathetic nerve activity by suppressing GABA_A inhibitory neuronal transmission.”

8. Lines 284-286: can’t the authors make the more explicit conclusion that it is the miR-33 in some population of catecholamine neurons that provides the support of BAT thermogenesis? Further, in lines 348-350, can the authors explicitly state that their GABAA-R subunit genetic data supports the conclusion that miR-33 acts to decrease the level of GABAA-R in these catecholamine neurons?

Thank you very much for these kind comments. We revised lines 284–286 as follows.

·Lines 320-322 in the revised manuscript;

“These data indicate that loss of miR-33 in catecholamine neurons is responsible for the impaired cold-induced thermogenesis and BAT activation, which contribute to obesity formation.”

We also revised Lines 348–350 as follows.

·Lines 383–385 in the revised manuscript;

“Thus, miR-33 promotes the activation of catecholamine neurons by decreasing the inhibitory effect of GABAergic inputs via down-regulation of *Gabrb2* and *Gabra4*, subunit-coding genes of GABA_A receptor.”

9. Lines 392-396: The preceding part of this paragraph makes no mention of a potential catecholamine excitatory input to DMH, which could be a site of miR-33 action to support BAT activation. The simple finding that Cre recombinase is found in the DMH of DBH-Cre mice does not logically indicate that “miR-33 in the DMH can enhance sympathetic nerve activity”, unless there are catecholamine neurons in the DMH that project to rRPa. This concluding sentence should be rewritten.

Shaun F. Morrison

Thank you very much for these critical comments. We revised the Lines 392–396 as follows.

•Lines 427–430 in the revised manuscript;

“Therefore, miR-33 in some populations of catecholamine-producing neurons can enhance sympathetic nerve activity by suppressing their upper thermoregulatory GABAergic inhibitory neurotransmission by targeting *Gabrb2* and *Gabra4*.”

Reviewers' comments, second round of review::

Reviewer #3 (Remarks to the Author):

Some of the questions from the previous review were satisfactorily addressed by the authors (i.e. quantification and improvement of Western blots). Others however, remain to be problematic. For example, one of the main criticisms in my original review was that data showing UCP1 changes were not quantified. Unfortunately, they were unable to show any differences in UCP1 expression when miR-33 is exerting its effect and the added immunohistochemistry staining does not show any convincing differences. Because it takes 30h for effects of miR-33 to be observed on UCP1 expression, likely this is a secondary effect.

I have some issues with the conclusions based on the statistical tests used. Much of the new data (and some of the previous) is done with a Mann-Whitney test, which is a non-parametric test that assumes a non-normal distribution. However, the data in the figures as presented appears to be normally distributed (at least visually) and no discussions of normality testing are present. The data need to be reanalyzed with a One-Way ANOVA (for multiple groups) or parametric t-test for 2 groups.

There are also issues with achieving statistical significance by simply increasing the number of mice per group in some measures, while concluding there are no statistically significant differences between groups with very small number of mice. Were the newly added mice (usually 1-2 per group) from a new cohort/litter of animals?

The ER stress data remains an issue, as it is poorly linked to the mechanism and observation in vivo. In response to the previous criticisms raised by all reviewers, the authors only show evidence in vitro of ER stress upon thapsigargin treatment, and not in vivo like was requested.

Reviewer #4 (Remarks to the Author):

The authors have performed many additional experiments and data analyses and have made extensive revisions to their text and figures. Earlier concerns have been satisfactorily addressed in this revision.

Reviewer #5 (Remarks to the Author):

In this extensive study, Horie et al. further address the role of miRNA-33 in adaptive thermogenesis, unveiling a pathway involving (putatively) central regulation of SNS projections to BAT. In their revision, the authors have considered all the comments and suggestions made by the initial referees, and have extensively revised their work, adding new pieces of data or reanalyzing (and expanding) previous datasets.

While these revisions have improved and completed the initial submission, there are some issues that need to be further considered or addressed in more detail. Many of them can be considered minor, although some conceptual issues are also in need for consideration:

1. It was previously mentioned in the comments of various referees that quantification of protein data was missing. While the authors have included some graphs with the quantification of WB, it is the impression of this referee that Western blot images in figures 1, 3, 5, and 6 are not accompanied by a graphical presentation of protein expression for each of the targets analyzed. In their responses, the authors mention that they have quantified and plotted protein data e.g., in Figure 1f, but I cannot find this graph in the revised MS. Please revise and include all quantifications, as requested.

2. As mentioned by some of the original referees, the characterization of the phenotype of mice with conditional ablation of miR-33 in DBH positive cells is the most salient and novel finding of the study, for which additional analyses and data were requested. I fully agree on the relevance of this dataset. Hence, it is somewhat surprising that most of this novel piece of data has been placed in Supplemental Figures, rather than regular figures. This should be reconsidered.

3. While combination of loss and gain of function studies reinforces the strength of the conclusions, the responses provided by the authors does not fully solve the potential limitation of the lack of cellular resolution of some the data (e.g., regarding ER stress), which needs to be at the very least better acknowledged in the paper. More importantly, the discordance of the time course for changes in miR-33 levels and ER stress markers induction after cold exposure is not solved in the revised MS, despite it is mentioned by several referees. The authors argue about the possibility of a mild ER stress in conditions of cold exposure, but the possibility that miR-33 is not actually involved in this phenomenon remains plausible, considering that induction of this miRNA happens at later time-points. In different forms, these weaknesses are pointed out by several referees. My suggestion is that the section of ER stress is better discussed or even that the ER stress connection is de-emphasized in face of the actual dataset.

4. I concur with referee-1 that some of the claims for the overarching implications of the data are speculative and too strong on the basis of the experimental data. In this context, I would suggest the authors tone down some statements regarding e.g. connection between atherosclerosis, SNS and miR-33 (see lines 464-467).

5. Further style editing is needed and in some places grammar and style typos are found.

6. Minor point, referee-3: Switch from 45% to 60% HFD in acute experiments. This is explained to the referee, but some brief justification should be provided in the text of the revised paper, to avoid any misleading interpretation by the reader.

REVIEWER COMMENTS

Reviewer #3 (Remarks to the Author):

We are grateful to Reviewer #3 for the important comments on our revised manuscript. As indicated in the following, we have taken all these comments and suggestions into account in the revised version of our manuscript.

[1] Some of the questions from the previous review were satisfactorily addressed by the authors (i.e. quantification and improvement of Western blots). Others however, remain to be problematic. For example, one of the main criticisms in my original review was that data showing UCP1 changes were not quantified. Unfortunately, they were unable to show any differences in UCP1 expression when miR-33 is exerting its effect and the added immunohistochemistry staining does not show any convincing differences. Because it takes 30h for effects of miR-33 to be observed on UCP1 expression, likely this is a secondary effect.

Thank you for your further comments about UCP1 expression in BAT. In fact, we found that mRNA of *Ucp1* was significantly increased when comparing between miR-33^{+/+} and miR-33^{-/-} mice after cold exposure. (Fig.1d and Fig. 3a). We added magnified pictures of immunostaining of BAT against UCP1 in Supplementary Fig. 3e. These pictures indicate dark cytoplasmic staining of UCP1 in miR-33^{+/+} mice. We further checked the UCP1 levels by Western blotting at 12h and 18h after cold exposure (earlier time points than 30 h) and found that the difference in UCP1 expression could be observed at 12h, and was more evident at 18h after cold exposure. These data suggest that the changes in UCP1 are likely to be primary effects of the miR-33 gene.

[2] I have some issues with the conclusions based on the statistical tests used. Much of the new data (and some of the previous) is done with a Mann-Whitney test, which is a non-parametric test that assumes a non-normal distribution. However, the data in the figures as presented appears to be normally distributed (at least visually) and no discussions of normality testing are present. The data need to be reanalyzed with a One-Way ANOVA (for multiple groups) or parametric t-test for 2 groups.

Thank you for these comments regarding statistics. We performed a Mann-Whitney (MW) test for the comparison of 2 groups and one-way ANOVA for the comparison of multiple groups. Per the reviewer's suggestion, we performed a D'Agostino-Pearson omnibus normality test with the data of more than 8 samples. We then gathered all the datasets in which all of the data were normally distributed from all figures, including supplementary files, and reanalyzed these datasets by unpaired t-tests. As a result of these analyses, almost all the datasets showed significant differences similar to those shown with the MW tests. There were only a few exceptions, which we detail below. Therefore, we consider the fundamental results described in our manuscript to be robust.

Fig. 2c, in dark phase; $p = 0.053$ for MW test, $p < 0.05$ for unpaired t-test.

Fig. 3e, Pparg; $p < 0.05$ for MW test, $p = 0.059$ for unpaired t-test.

Fig. 5g, Pparg; $p < 0.05$ for MW test, $p = 0.066$ for unpaired t-test.

Fig. 7h, Adrb3 shControl; $p = 0.087$ for MW test, $p < 0.05$ for unpaired t-test.

[3] There are also issues with achieving statistical significance by simply increasing the number of mice per group in some measures, while concluding there are no statistically significance differences between

groups with very small number of mice. Were the newly added mice (usually 1-2 per group) from a new cohort/litter of animals?

Yes, newly added mice were from a new litter of mice.

[4] The ER stress data remains an issue, as it is poorly linked to the mechanism and observation in vivo. In response to the previous criticisms raised by all reviewers, the authors only show evidence in vitro of ER stress upon thapsigargin treatment, and not in vivo like was requested.

Thank you for the important comment. Because it takes several days for hypothalamic miR-33 to increase in response to cold exposure, this may indicate an adaptive response to cold environments and contribute to further increases in BAT activity. We consider this response to be a subacute effect rather than an acute effect. Our genetic data indicate that hypothalamic miR-33 levels affect an acute response to cold exposure. Further alternation of hypothalamic miR-33 levels can affect BAT activity in a subacute manner. However, this induction of miR-33 is relatively mild and takes time. Thus it is considered to be rather a complementary or augmentation response. We agree that the in vivo data regarding ER stress is limited, and there are some weak points in the hypothesis that cold stress induces miR-33 via ER stress. Therefore, we decided to place these data in Supplementary files. We also revised the text to de-emphasize this by deleting some sentences and adding some thoughts about limitations in the discussion.

•Lines 458-468 in the revised manuscript;

“The difference in the duration of miR-33 elevation between our *in vitro* and *in vivo* results may be due to the degree of ER stress, although further *in vivo* analysis is surely required. It takes several days for hypothalamic miR-33 to increase after cold exposure, which may be an adaptive or complementary response to cold environments. Because miR-33 in the brain is induced by stresses such as fear conditioning, behavior stress, and cold stress, and given the ability of miR-33 to increase sympathetic nerve tone by suppressing GABAergic inhibitory transmission, this machinery is likely to be a physiological adaptive defense mechanism against life-threatening stress. However, the induction of miR-33 is relatively mild and takes time. In general, our results

support that miR-33 is important for inducing sympathetic nerve activation, and suggest that its induction may be a complement to this neural mechanism under cold exposure or HFD feeding.”

We also revised some other related sentences in the main text as follows.

- Line 41 in this revised manuscript; we deleted “via ER stress” in the sentence of the abstract.

“We found that cold stress increased miR-33 levels in the hypothalamus,”

We deleted the following sentence in Lines 110-111 in the previous version of our manuscript;

“In wild-type mice, cold stress up-regulated the expression of miR-33 in the hypothalamus via endoplasmic reticulum (ER) stress.”

- Line 241 in this revised manuscript; we added “possibly” as follows.

“Cold exposure induces hypothalamic miR-33, possibly via ER stress”

- Lines 260-261 in this revised manuscript; we reworded “sufficient” to “important” as follows.

“Because the miR-33 increase was prominent at a low concentration, mild ER stress may be important for miR-33 induction.”

- Lines 263-264 in this revised manuscript; we added “may” as follows.

“These results suggest that cold exposure and/or HFD feeding may induce miR-33 expression via increased ER stress in the hypothalamus.”

We deleted the following sentence in Lines 449-451 in the previous version of our manuscript;

“In fact, although a significant BIP induction was observed, an increase in cleaved-caspase 3 was not observed in this condition, which indicates that ER stress evoked by cold stress is mild.”

- Lines 520-523 in this revised manuscript; we deleted the following phrase, “(miR-33 in the brain), which is upregulated by ER stress,” and revised this sentence as follows.

“During cold stress, miR-33 in the brain (Dbh-positive neurons in particular) contributes to the maintenance of BAT thermogenesis and whole-body metabolism through increased sympathetic nerve activity by suppressing GABA_A inhibitory neuronal transmission.”

·Lines 248 and 455 in this revised manuscript; we additionally cited another report which showed that ER stress induces miR-33 in astrocytes (Brain Behav Immun 46, 270-279, 2015; ref. 32).

Reviewer #4 (Remarks to the Author):

The authors have performed many additional experiments and data analyses and have made extensive revisions to their text and figures. Earlier concerns have been satisfactorily addressed in this revision.

We appreciate the work of Reviewer #4 in reviewing our manuscripts. Thank you very much.

Reviewer #5 (Remarks to the Author):

We are grateful to Reviewer #5 for the important comments and useful suggestions that helped to improve the manuscript considerably. As indicated in the responses that follow, we have taken all these comments and suggestions into account in the revised version of our manuscript.

In this extensive study, Horie et al. further address the role of miRNA-33 in adaptative thermogenesis, unveiling a pathway involving (putatively) central regulation of SNS projections to BAT. In their revision, the authors have considered all the comments and suggestions made by the initial referees, and have extensively revised their work, adding new pieces of data or reanalyzing (and expanding) previous datasets.

While these revisions have improved and completed the initial submission, there are some issues that need to be further considered or addressed in more detail. Many of them can be considered minor, although some conceptual issues are also in need for consideration:

[1] It was previously mentioned in the comments of various referees that quantification of protein data was missing. While the authors have included some graphs with the quantification of WB, it is the impression of this referee that Western blot images in figures 1, 3, 5, and 6 are not accompanied by a graphical presentation of protein expression for each of the targets analyzed. In their responses, the author mention that they have quantified and plotted protein data e.g., in Figure 1f, but I cannot find this graph in the revised MS. Please revise and include all quantifications, as requested.

Thank you for the comment about quantifying WB. We did quantify the WB results, and presented these data in Supplementary files in the previous version. In the current revision, quantification of Fig.1f is presented in S3d, Fig. 3f is in S5d, Fig. 4e is in S7b, Fig. 6e is in S8g, S2c is in S2d, S6c is in S6d, and S6f is in S6g.

[2] As mentioned by some of the original referees, the characterization of the phenotype of mice with conditional ablation of miR-33 in DBH positive cells is the most salient and novel finding of the study, for which additional analyses and data were requested. I fully agree on the relevance of this dataset. Hence, it is somewhat surprising that most of this novel piece of data has been placed in Supplemental Figures, rather than regular figures. This should be reconsidered.

Thank you for the positive comment, and for the suggestion regarding the data from miR-33^{ff} DBH-Cre mice. We agree completely, and in the current version, we present these data in Fig. 5 and Fig. 6 (which were Supplementary files in the previous version).

[3] While combination of loss and gain of function studies reinforces the strength of the conclusions, the responses provided by the authors does not fully solve the potential limitation of the lack of cellular resolution of some the data (e.g., regarding ER stress), which needs to be at the very least better acknowledged in the paper. More importantly, the discordance of the time course for changes in miR-33 levels and ER stress markers

induction after cold exposure is not solved in the revised MS, despite it is mentioned by several referees. The authors argue about the possibility of a mild ER stress in conditions of cold exposure, but the possibility that miR-33 is not actually involved in this phenomenon remains plausible, considering that induction of this miRNA happens at later time-points. In different forms, these weaknesses are pointed out by several referees. My suggestion is that the section of ER stress is better discussed or even that the ER stress connection is de-emphasized in face of the actual dataset.

Thank you for this important comment. We agree that the data on ER stress, especially in vivo, is limited, and there is some weakness to the hypothesis that cold stress induces miR-33 via ER stress. According to the reviewer's suggestion, in the revised text we placed these data in Supplementary files to de-emphasize the section concerning ER stress. Moreover, we deleted the phrase "via ER stress" in the abstract, deleted some other related sentences in the main text, and added some text about these limitations in the discussion.

•Line 41 in this revised manuscript; we deleted "via ER stress" in the sentence of the abstract.

"We found that cold stress increased miR-33 levels in the hypothalamus,"

We deleted the following sentence in Lines 110-111 in the previous version of our manuscript;

"In wild-type mice, cold stress up-regulated the expression of miR-33 in the hypothalamus via endoplasmic reticulum (ER) stress."

•Line 241 in this revised manuscript; we added "possibly" as follows.

"Cold exposure induces hypothalamic miR-33, possibly via ER stress"

•Lines 260-261 in this revised manuscript; we reworded "sufficient" to "important" as follows.

"Because the miR-33 increase was prominent at a low concentration, mild ER stress may be important for miR-33 induction."

•Lines 263-264 in this revised manuscript; we added "may" as follows.

"These results suggest that cold exposure and/or HFD feeding may induce

miR-33 expression via increased ER stress in the hypothalamus.”

We deleted the following sentence in Lines 449-451 in the previous version of our manuscript;

“In fact, although a significant BIP induction was observed, an increase in cleaved-caspase 3 was not observed in this condition, which indicates that ER stress evoked by cold stress is mild.”

·Lines 248 and 455 in this revised manuscript; we additionally cited another report which showed that ER stress induces miR-33 in astrocytes (Brain Behav Immun 46, 270-279, 2015; ref.32).

·Lines 458-468 in this revised manuscript; we deleted some sentences and added some text about limitation as follows.

“The difference in the duration of miR-33 elevation between our *in vitro* and *in vivo* results may be due to the degree of ER stress, although further *in vivo* analysis is surely required. It takes several days for hypothalamic miR-33 to increase after cold exposure, which may be an adaptive or complementary response to cold environments. Because miR-33 in the brain is induced by stresses such as fear conditioning, behavior stress, and cold stress, and given the ability of miR-33 to increase sympathetic nerve tone by suppressing GABAergic inhibitory transmission, this machinery is likely to be a physiological adaptive defense mechanism against life-threatening stress. However, the induction of miR-33 is relatively mild and takes time. In general, our results support that miR-33 is important for inducing sympathetic nerve activation, and suggest that its induction may be a complement to this neural mechanism under cold exposure or HFD feeding.”

·Lines 520-523 in this revised manuscript; we deleted the following phrase, “(miR-33 in the brain), which is upregulated by ER stress,” and revised this sentence as follows.

“During cold stress, miR-33 in the brain (Dbh-positive neurons in particular) contributes to the maintenance of BAT thermogenesis and whole-body metabolism through increased sympathetic nerve activity by suppressing GABA_A inhibitory neuronal transmission.”

[4] I concur with referee-1 that some of the claims for the overarching implications of the data are speculative and too strong on the basis of the experimental data. In this context, I would suggest the authors tone down some statements regarding e.g. connection between atherosclerosis, SNS and miR-33 (see lines 464-467).

We appreciate and agree with the reviewer's comment that our previous discussion of connections between SNS, atherosclerosis, and miR-33 was too strong given the actual data, and we have toned down the sentence in 464-467 accordingly.

•Lines 478-480 in this revised manuscript (Lines 464-467 in the previous version); we added "can" in this sentence.

"Because increased sympathetic nerve activity is known to accelerate atherosclerosis by several mechanisms⁵¹⁻⁵⁴, its reduction through deletion or inhibition of miR-33 can contribute to the amelioration of atherosclerosis."

[5] Further style editing is needed and in some places grammar and style typos are found.

Thank you for pointing out grammar and style typos. We have re-submitted our manuscript for English proofreading, and have double-checked the manuscript ourselves.

[6] Minor point, referee-3: Switch from 45% to 60% HFD in acute experiments. This is explained to the referee, but some brief justification should be provided in the text of the revised paper, to avoid any misleading interpretation by the reader.

Thank you for pointing out this unclear description. According to this suggestion, we have added an explanation about why we switched from 45% to 60% HFD in the short-term feeding in the revised text.

•Line134-136 in this revised manuscript;

“To understand the direct effects of HFD feeding, we conducted a shorter-term study lasting 2 weeks, during which mice were fed a diet with higher fat content (60%) to increase the burden for a short time.”

Reviewers' comments, third round of review::

Reviewer #3 (Remarks to the Author):

The authors addressed my concerns, although I recommend review by a statistical reviewer to ensure the tests used were appropriate.

Reviewer #5 (Remarks to the Author):

The authors have efficiently addressed all the major comments made in my previous evaluation. I have just one minor additional comment. I fully understand that the regular figures are a bit over-crowded but I find a bit complex to follow how quantification of WB data is presented in the revised paper. In its present form, quantitative data of WB shown in regular figures is displayed in supplemental figures, with the following distribution:

"In the current revision, quantification of Fig.1f is presented in S3d, Fig. 3f is in S5d, Fig. 4e is in S7b, Fig. 6e is in S8g, S2c is in S2d, S6c is in S6d, and S6f is in S6g."

I personally feel this is a bit cumbersome, but I defer to the criterion of the editor on whether or not it might be better to display quantitative data in the same figures where the WB are presented.

Otherwise, I am satisfied with the revisions made by the authors.

Response to Reviewer #3

We are grateful to Reviewer #3 for the useful comment.

The authors addressed my concerns, although I recommend review by a statistical reviewer to ensure the tests used were appropriate.

Thank you very much for the valuable comment. We provide the raw data underlying all graphs including statistics tests, and uncropped version of blots as a Source Data for a statistical review. We appreciate the work of Reviewer #3 in reviewing our manuscripts.

Response to Reviewer #5

We are grateful to Reviewer #5 for the useful suggestion and comment about the presentation of the quantification data for WB.

The authors have efficiently addressed all the major comments made in my previous evaluation.

I have just one minor additional comment. I fully understand that the regular figures are a bit over-crowded but I find a bit complex to follow how quantification of WB data is presented in the revised paper. In its present form, quantitative data of WB shown in regular figures is displayed in supplemental figures, with the following distribution: “In the current revision, quantification of Fig.1f is presented in S3d, Fig. 3f is in S5d, Fig. 4e is in S7b, Fig. 6e is in S8g, S2c is in S2d, S6c is in S6d, and S6f is in S6g.”

I personally feel this is a bit cumbersome, but I defer to the criterion of the editor on whether or not it might be better to display quantitative data in the same figures where the WB are presented.

Otherwise, I am satisfied with the revisions made by the authors.

We fully agree with this reviewer’s comment. Therefore, we placed these data on the same figure of the blotting data, as follows. Quantification of Fig.1f is placed in Fig.1g, Fig. 3f in Fig. 3g, Fig. 4e in Fig. 4f, and Fig.6e in Fig. 6f in the revised manuscript. We appreciate the work of Reviewer #5 in reviewing our manuscripts.

Reviewers' comments, fourth round of review::

Reviewer #6 (Remarks to the Author):

Page 5, line 152: it states that "we found that rectal temperature decreased faster in miR-33^{-/-} mice than in miR-33^{+/+}." According to figure 1a, a series of pointwise two-sample Mann-Whitney tests were conducted to test significance of between-group difference at each time point, rather than test whether the rate of decline in body temperature was different. To make a direct inference on the latter, a repeated measures ANOVA is more appropriate.

Figure 1d, Figure 3a, e, Figure 5g, Figure 7a, Figure S2a, etc. regarding the quantitative PCR analysis of thermogenic genes, it needs to be justified why multiple comparison adjustment is not needed here to account for significance by chance due to the simultaneous testing of multiple genes.

Regarding the analyses of oxygen consumption data (e.g., Fig. 2a, 2b), the test of between-group difference was based on individual mean oxygen consumption rate during the light and dark phase. The authors are encouraged to use functional data analysis for a more rigorous and powerful testing of the differences between the curves (e.g., use fdANOVA in R).

Reviewer #6 (Remarks to the Author):

We are grateful to Reviewer #6 for the helpful comments and suggestions regarding the statistics on our revised manuscript. As indicated in the following, we have taken all these comments and suggestions into account in the revised version of our manuscript.

Page 5, line 152: it states that “we found that rectal temperature decreased faster in miR-33^{-/-} mice than in miR-33^{+/+}.” According to figure 1a, a series of pointwise two-sample Mann–Whitney tests were conducted to test significance of between-group difference at each time point, rather than test whether the rate of decline in body temperature was different. To make a direct inference on the latter, a repeated measures ANOVA is more appropriate.

Thank you very much for the useful suggestion about the statistics for rectal temperature. Following this suggestion, we re-analyzed the data on rectal temperature using two-way repeated measures ANOVA for Fig. 1a, Fig. 5d, Fig. 7d, Fig. 7e, Fig. 7f, Fig. S7j, and Fig. S10h. All results were similar to the previous ones. We revised the manuscript in accordance with these new analyses.

Figure 1d, Figure 3a, e, Figure 5g, Figure 7a, Figure S2a, etc. regarding the quantitative PCR analysis of thermogenic genes, it needs to be justified why multiple comparison adjustment is not needed here to account for significance by chance due to the simultaneous testing of multiple genes.

Thank you very much for pointing out this issue. In this study, we preselected specific thermogenic genes that have been shown to play important roles in BAT biology from previous studies, and measured their expression levels. We did not randomly measure multiple genes or intentionally pick several genes that showed statistical significance. Therefore, we consider that multiple comparison adjustment for multiple gene measurement is not needed in this case, although it is necessary for analysis of microarray or RNA-seq where genes are randomly selected after measurement and comparison of all genes.

We also checked recent articles, in which quantitative PCR analysis of multiple genes was simultaneously performed for BAT analysis. We could not find adjustments to quantitative PCR analysis for multiple gene measurements. Therefore, the adjustment for quantitative PCR analysis in such cases does not seem to be common. The followings are examples of

articles.

- Tongyu Liu, et al. BAF60a deficiency uncouples chromatin accessibility and cold sensitivity from white fat browning. *Nature Communications* 11, 2379 (2020).
- Alexander J. Knights, et al. Eosinophil function in adipose tissue is regulated by Krüppel-like factor 3 (KLF3). *Nature Communications* 11, 2922 (2020).
- Diana Teh Chee Siang, et al. The RNA-binding protein HuR is a negative regulator in adipogenesis. *Nature Communications* 11, 213 (2020).
- Fabrizio C. Lucchini, et al. ASK1 inhibits browning of white adipose tissue in obesity. *Nature Communications* 11, 1642 (2020).
- Hanying Ding, et al. Fasting induces a subcutaneous-to-visceral fat switch mediated by microRNA-149-3p and suppression of PRDM16. *Nature Communications* 7, 11533 (2016).
- Serkan Kir, et al. Tumour-derived PTH-related protein triggers adipose tissue browning and cancer cachexia. *Nature* 513, 100-104 (2014).
- Andrea Galmozzi, et al. PGRMC2 is an intracellular haem chaperone critical for adipocyte function. *Nature* 576, 138-142 (2019).

Regarding the analyses of oxygen consumption data (e.g., Fig. 2a, 2b), the test of between-group difference was based on individual mean oxygen consumption rate during the light and dark phase. The authors are encouraged to use functional data analysis for a more rigorous and powerful testing of the differences between the curves (e.g., use fdANOVA in R).

Thank you very much for the useful comment and suggestion about functional data analysis for the oxygen consumption curve. Following the suggestion, we performed fdANOVA analysis using R software on the oxygen consumption curve.

As a result, the p-value between the curves was $p=0.009$ in Fig. 2a and $p=0.01$ in Fig. 2b.

We also analyzed other curves and the results were as follows.

Fig. 2g; $p=0.031$, Fig. 2h; $p=0.027$, Fig. 4h; $p=0.012$, Fig. 6a; $p=0.096$, S1e; $p=0.005$, S3f; $p=0.063$, S7f; $p=0.304$, S7h; $p=0.022$.

We revised the manuscript in accordance with these new analyses.

- Tomasz Górecki and Łukasz Smaga. fdANOVA: an R software package for analysis of variance for univariate and multivariate functional data. *Computational Statistics* 34, 571-597 (2019).

No further comments.